# ClimaQA: An Automated Evaluation Framework for Climate Question Answering Models

**Veeramakali Vignesh Manivannan, Yasaman Jafari, Srikar Eranky, Spencer Ho,**
**Rose Yu, Duncan Watson-Parris, Yian Ma, Leon Bergen & Taylor Berg-Kirkpatrick** [*]
University of California, San Diego
{vmanivannan,yajafari,seranky,s8ho,roseyu,
dwatsonparris,yianma,lbergen,tberg}@ucsd.edu

## Abstract

The use of Large Language Models (LLMs) in climate science has recently gained significant attention. However, a critical issue remains: the lack of a comprehensive evaluation framework capable of assessing the quality and scientific validity of model outputs. To address this issue, we develop *ClimaGen* (Climate QA Generator), an adaptive learning framework that generates question-answer pairs from graduate textbooks with climate scientists in the loop. As a result, we present *ClimaQA-Gold*, an expert-annotated benchmark dataset alongside *ClimaQA-Silver*, a large-scale, comprehensive synthetic QA dataset for climate science. Finally, we develop evaluation strategies and compare different LLMs on our benchmarks. Our results offer novel insights into various approaches used to enhance knowledge of climate LLMs. ClimaQA's source code is publicly available at https://github.com/Rose-STL-Lab/genie-climaqa

## 1 Introduction

Climate change is one of the most pressing global challenges today, with profound impacts on ecosystems, economies, and societies. In recent years, Large Language Models (LLMs) have gained significant interest in climate science (Thulke et al., 2024; Nguyen et al., 2024; Cao et al., 2024) due to their potential to transform climate predictions and enable applications in climate policy analysis, environmental decision-making, and public education. By improving LLMs' understanding of climate science, we can empower stakeholders to make informed decisions, develop actionable solutions, and foster broader awareness of climate issues. However, while LLMs are powerful, they often fall short when it comes to answering technical questions requiring high precision such as *What is the net effect of Arctic stratus clouds on the Arctic climate?* Even advanced models like GPT-4 exhibit epistemological inaccuracies in Climate Question-Answering (QA) tasks (Bulian et al., 2024), raising concerns about their reliability in scientific workflows.

This highlights the need for a domain-specific evaluation framework to assess the quality and validity of outputs generated by these models. Current benchmarks for Large Language Models (LLMs) predominantly focus on linguistic accuracy or general factual correctness (Bai & Wang, 2021), but they fail to address the unique demands of climate science, where factual rigor, domain-specific knowledge, and robust reasoning are essential. Although some work has explored the scientific evaluation of LLMs (Table 1), they either rely heavily on manual expert input or employ fully synthetic question generations. To address this issue, we develop **ClimaGen**, an adaptive learning framework for creating benchmarks in collaboration with domain experts to evaluate scientific question-answering models, specifically for climate science but adaptable to other scientific disciplines, shown in Figure 1. This enables us to achieve a balance between utilizing the efficiency of LLMs and the expertise of domain specialists.

Using our framework, we introduce a novel benchmark for evaluating question-answering models in climate science across three scientific QA task forms: multiple-choice, freeform, and cloze. The questions are designed with varying levels of complexity, challenging the models to demonstrate a range of reasoning abilities from basic factual recall to scientific reasoning and scenario applications.

---

[*]correspondence to Rose Yu & Taylor Berg-Kirkpatrick

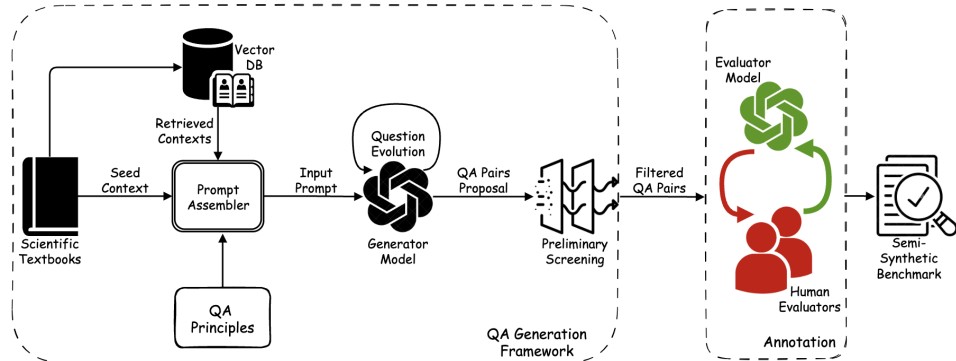

Figure 1: ClimaGen - Our proposed Automated Benchmark Creation Framework. The QA generation framework creates synthetic data from seed contexts extracted from graduate-level textbooks using LLMs to generate base-level question-answer pairs and evolve them by adding complexities to the same. These are validated by domain experts during the annotation process to produce the semi-synthetic benchmark. The evaluator model is trained actively using the human-labeled examples in order to completely automate the process.

The benchmark consists of two datasets: **ClimaQA-Gold** – an expert-annotated dataset with a total of 566 questions validated by climate scientists, ensuring high scientific rigor, and **ClimaQA-Silver** – a large-scale synthetic dataset consisting of 3000 questions generated by our framework, providing substantial ground truth data for model fine-tuning at scale. Together, these datasets enable comprehensive performance assessment of LLMs in climate science, specifically for scientific QA tasks. We evaluate several LLMs on our benchmark under different settings. We observe that most models struggle with reasoning-based multiple-choice questions (MCQs) and Retrieval Augmented Generation (RAG) (Lewis et al., 2020) significantly outperforms Continued Pre-training and Supervised Fine-tuning across different tasks.

In summary, our contributions are as follows:

- Creation of publicly releasable datasets: expert-annotated (ClimaQA-Gold) and synthetic (ClimaQA-Silver), along with tailored evaluation metrics facilitating both rigorous assessment and large-scale fine-tuning on 3 scientific QA task forms: multiple-choice, freeform, and cloze, with varying levels of complexity.

- Development of a generalized adaptive learning framework (ClimaGen) for creating scientific benchmarks at scale in collaboration with domain experts for evaluation of natural language question-answering models on scientific accuracy.

- Evaluation of state-of-the-art LLMs on climate science QA tasks, with insights into improving scientific accuracy.

Table 1: Comparison of scientific benchmarks. Automated indicates automatic creation, Validated shows expert validation, Multi-task represents multiple task types, and Multi-level represents questions of varying complexity

| Dataset | Domain | Source | Size | Automated | Validated | Multi-Task | Multi-Level |
|---------|--------|--------|------|-----------|-----------|------------|-------------|
| ScienceQA | Science | Hi-Scl Text | 21000 | ✗ | ✓ | ✗ | ✗ |
| Pira2 | Ocean | Research | 2250 | ✗ | ✓ | ✓ | ✗ |
| SciQA | Comp Sci | ORKG | 2500 | ✓ | ✓ | ✗ | ✗ |
| Climate Crisis | Climate | None | 20000 | ✓ | ✗ | ✗ | ✗ |
| SciQAG-24D | Science | Research | 8531 | ✓ | ✗ | ✗ | ✗ |
| *ClimaQA-Gold* | Climate | Grad Text | 566 | ✓ | ✓ | ✓ | ✓ |
| *ClimaQA-Silver* | Climate | Grad Text | 3000 | ✓ | ✗ | ✓ | ✓ |

## 2   Related Work

ScienceQA (Lu et al., 2022) contains a vast collection of multimodal MCQs manually curated from high school textbooks. Pira2 (Pirozelli et al., 2024) consists of expert-created questions derived from research articles focused on oceans, the Brazilian coast, and climate change. The creation of these benchmarks required substantial manual effort. SciQA (Auer et al., 2023) innovatively generates freeform QA pairs by leveraging hand-crafted queries on the Open Research Knowledge Graph (Jaradeh et al., 2019) primarily drawing from computer science literature. Although these pairs are factually accurate, they do not include an automatic evaluation method for generated responses. Climate Crisis QA (Zhu & Tiwari, 2024) and SciQAG-24D (Wan et al., 2024) explore synthetic data generations using Large Language models. However, their approaches are prone to suffer from hallucinations and lack of scientific validity. To address this, we introduce a gold-standard dataset, rigorously validated by domain experts, alongside a large-scale silver dataset whose generation process was guided by these expert validation labels. Moreover, existing benchmarks generally focus on a single QA format and lack scientifically aligned evaluation metrics. Our benchmark contains questions of three distinct scientific QA task forms at varying levels of complexity, along with evaluation metrics tailored to them. Table 1 presents a comparison of various scientific benchmarks.

Previous work on automated MCQ generation has focused on selecting keywords and generating distractors based on contextual information. (A. Nwafor & E. Onyenwe, 2021)) utilize traditional NLP techniques, such as TF-IDF, for keyword extraction, while (Mehta et al., 2021) leverage BERT (Devlin, 2018) for summarization and keyword extraction, employing WordNet (Miller, 1995) to generate distractors. (Das et al., 2021) applies RAKE (Rose et al., 2010) for keyword extraction and clustering methods for distractor generation. Other approaches, such as utilizing dependency trees (Afzal & Mitkov, 2014), have also been explored for MCQ creation. These methods typically focus on generating MCQs with single-word answers. However, the recent advancements in LLMs have enabled the creation of more complex MCQs with longer, detailed answer choices. (Meißner et al., 2024) demonstrate the automated generation of self-assessment quizzes using LLMs, while (Hang et al., 2024) explore self-refining prompting techniques for improved MCQ generation. Recent studies, including (Olney, 2023) and (Doughty et al., 2024), suggest that LLMs can generate MCQs comparable to those created by humans, though (Grévisse et al., 2024) emphasize the importance of human oversight to ensure the quality and pedagogical relevance of these questions.

## 3   ClimaQA - Climate Question Answering Benchmark

The ClimaQA benchmark is built on questions generated from graduate-level climate science textbooks, ensuring alignment with the precise terminology and complex theories of the field. These textbooks provide a reliable source for generating both the expert-validated ClimaQA-Gold dataset and the synthetic ClimaQA-Silver dataset. By leveraging textbook content and combining it with expert review, ClimaQA facilitates rigorous evaluation and fine-tuning of LLMs across freeform, multiple-choice, and cloze question-answering tasks in climate science. Our expert-validated dataset, ClimaQA-Gold, ensures that the evaluation questions are accurate, relevant, and reflect the current understanding of climate science.

### 3.1   Scientific Question Answering

To thoroughly evaluate a model's ability to handle scientific questions, we create our benchmark dataset to focus on the different complexities of scientific reasoning. The aim is to test the model's ability to engage with scientific concepts at different levels of understanding and scenario application.

Our benchmark consists of questions of three levels of complexity. The first level involves basic questions designed to test straightforward factual understanding. The second level introduces reasoning, requiring the model to connect multiple scientific facts or principles. The third level involves hypothetical scenarios, testing the model's ability to apply scientific knowledge in unseen contexts. These questions challenge the model's scientific reasoning in different ways, from knowledge recall to advanced reasoning and problem-solving in dynamic contexts. A question from each level of complexity is shown in Figure 2 as an example.

```
Base

Question - What is a crucial factor to ensure when collecting data for calibration purposes?
Options -
    a) Using different solution sources for each data set.
    b) Consistency in equipment setup and data collection procedures.
    c) Changing the calibration locations frequently to avoid bias.
    d) Varying the nebuliser type for each calibration date.
Answer - b

------------------------------------------------------------------------------------------
Reasoning

Question - Why is consistency in equipment setup and data collection procedures considered a
crucial factor for collecting data for calibration purposes?
Options -
    a) It ensures that the calibration process is completed faster.
    b) It helps in minimizing errors and maintaining reliable and accurate measurements.
    c) It helps in identifying outliers in the data sets more effectively.
    d) It allows for easy integration of new equipment without affecting the calibration
       results.
Answer - b

------------------------------------------------------------------------------------------
Hypothetical Scenario

Question - How might the calibration accuracy be affected if the driers and DMA were changed
between different calibration sessions?
Options -
    a) The calibration accuracy would deteriorate due to inconsistent conditions.
    b) The calibration accuracy would fluctuate depending on the type of nebuliser used.
    c) The calibration accuracy would remain unaffected by the change in equipment.
    d) The calibration accuracy would improve due to the variability introduced.
Answer - a
```

Figure 2: Examples of Question Evolution. The first is the initial version of the generated question. The second is the enhanced version of the question that requires scientific reasoning to answer. The third is the modified version of the question that involves a hypothetical scenario. The contexts from the textbook data were used during the question evolution.

The questions come in three different task forms, demonstrated in Figure 3:

- **MCQ:** The model selects correct answers from predefined options, assessing its factual accuracy and decision-making under constrained conditions.
- **Freeform:** The model generates detailed, structured responses, testing its ability to reason logically and produce scientifically sound explanations.
- **Cloze:** The model fills in blanks with appropriate scientific terms, evaluating its contextual understanding and use of domain-specific vocabulary.

Together, the benchmark as shown in Table 2 provides a robust framework for evaluating an LLM's proficiency in scientific reasoning, critical thinking, and applying knowledge in unseen scenarios.

## 3.2 EVALUATION METRICS

Although assessing multiple-choice question-answering is relatively simple, the other two tasks present more challenges. To address this, we propose and validate the following evaluation metrics for freeform and cloze question-answering. A more detailed case study to demonstrate the robustness of these metrics can be found in the Appendix A.2.3. We use these metrics to report experimental results in Section 5

Table 2: Contents of the ClimaQA dataset. Both ClimaQA-Gold and ClimaQA-Silver include 3 task-forms with varying levels of complexity for MCQ and Freeform.

| Dataset | Task | Base | Reasoning | Hypothetical | Total |
|---|---|---|---|---|---|
| ClimaQA-Gold | MCQ | 126 | 72 | 47 | 245 |
| | Freeform | 54 | 52 | 55 | 161 |
| | Cloze | - | - | - | 160 |
| ClimaQA-Silver | MCQ | 501 | 264 | 235 | 1000 |
| | Freeform | 507 | 241 | 252 | 1000 |
| | Cloze | - | - | - | 1000 |

```
Free Form

Question - What is the purpose of measurements at monitoring sites in relation to air
pollutants and acid rain?
Answer - Measurements at monitoring sites were initially made to demonstrate long-range
transport of air pollutants and to inform policies to mitigate acid rain and associated
damage to ecosystems and infrastructure.

-------------------------------------------------------------------------------------
Multiple Choice

Question - Why is wet deposition more efficient when the aerosol concentration is high?
Options -
    a) Since high aerosol concentration reduces the time aerosols spend in the atmosphere.
    b) As there are more aerosol particles available for removal by precipitation processes.
    c) Due to the larger size of aerosol particles facilitating easier capture by
       precipitation.
    d) Because high aerosol concentration leads to increased collision and coagulation rates.
Answer - b

-------------------------------------------------------------------------------------
Cloze

Question - The thickness of the layer between any two pressure surfaces is proportional to
the mean virtual temperature of the layer, leading to a direct relationship between the
change in <blank> height and the thickness of the intervening layer.
Answer - geopotential
```

Figure 3: Examples of the three types of scientific question-answering tasks presented in our benchmark

### 3.2.1 FREEFORM QA

Various metrics are employed to evaluate sentence similarity, ranging from surface-level comparisons to deeper semantic analysis. Lexical metrics such as BLEU (Papineni et al., 2002), ROUGE (Lin, 2004), and METEOR (Banerjee & Lavie, 2005) focus on exact word or n-gram matching, rendering them useful for tasks where token overlap is crucial, such as machine translation. In contrast, semantic metrics like BERTScore (Zhang et al., 2019), Word Mover's Distance (WMD) (Huang et al., 2016), and Sentence-BERT (Reimers, 2019) are more advanced, capturing the meanings of sentences through embeddings. These metrics are better suited for tasks that necessitate an understanding of meaning, such as paraphrase detection. However, they may not adequately assess factual accuracy.

To measure the factual accuracy of generated answers relative to reference answers, we propose the use of a factual entailment classifier, reporting the confidence level as the factual accuracy score. Instruction-tuned models, such as GPT-4, have demonstrated superior performance on textual en-

tailment tasks and have shown the ability to generalize across various datasets (Sanyal et al., 2024). We employ the GPT-4o-mini model with the prompt below for factual entailment. This method achieved $81\%$ zero-shot classification accuracy on the Climate-Fever dataset (Diggelmann et al., 2020) indicating the ability to measure factual accuracy.

> You are a climate expert who annotates whether a given claim either SUPPORTS or REFUTES the presented evidence. You will be provided with the following input:
>
> **Evidence**: $\langle evidence \rangle$
> **Claim**: $\langle claim \rangle$
>
> Respond with only one word: SUPPORTS if the claim supports the evidence and REFUTES otherwise.

To use this for scoring freeform QA, the reference answer was used as the evidence and the generated answer was used as the claim. The confidence score was computed by applying sigmoid smoothing to the logit scores, with the temperature parameter set to $T = 5$. Note that the choice of $T$ does not alter the score trend; it was selected to optimally scale values between 0 and 1. If $l_s$ and $l_r$ are the logit scores for SUPPORTS and REFUTES respectively, then

$$\text{Factual Accuracy} = \text{SigmoidSmooth}\left(\frac{l_s}{l_s + l_r}\right)$$

Overall, we report three metrics for freeform question answering (QA): **BLEU**, **BERTScore**, and **Factual Accuracy** to evaluate different aspects of the generated answers.

### 3.2.2 CLOZE QA

Performance on this task is typically evaluated using the exact match metric. However, this approach has limitations due to the existence of multiple correct answers. A generated answer may differ from the reference answer while remaining contextually and semantically valid; for instance, while 'point' and 'temperature' are semantically distinct terms, they can be contextually similar within phrases like 'freezing point' and 'freezing temperature.' This illustrates that semantic relationships can depend heavily on context.

To address this challenge, we introduce a metric that captures the semantic similarity between the generated answer and the ground-truth answer for a more nuanced assessment of model performance. Specifically, we utilize a context window to extract two phrases: one with the $\langle$**blank**$\rangle$ replaced by the reference answer and the other with the generated answer. The semantic similarity is measured using cosine similarity between the Universal Sentence Encoder (Cer, 2018) embeddings of these phrases.

To evaluate the robustness of this approach, we synthetically generated 219 cloze questions with our framework (described in Section 4), which were answered by the GPT-4o-mini model. We collected 32 questions where the generated answers did not exactly match the reference answer. These answers were then labeled as *wrong* or *correct* by domain experts based on scientific and contextual accuracy. We plotted the average cosine similarities of phrases for these questions as shown in Figure 4, concluding that a context window of size 4

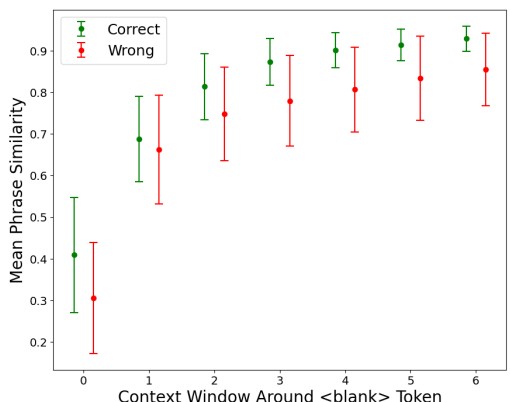

Figure 4: Mean Phrase Similarity for Correctly Answered and Incorrectly Answered Cloze Questions

most effectively differentiates between correct and incorrect answers. This configuration yields the maximum difference in scores while maintaining sufficiently high scores for correct answers. The cosine similarities were subsequently rescaled to emphasize these differences. If $e_1$ and $e_2$ are the embeddings of the respective phrases as mentioned above, then

$$\text{Phrase Similarity} = 2 \times (\text{CosineSimilarity}(e_1, e_2) - 0.5)$$

We report two metrics for cloze question answering (QA): **Exact Match** and **Phrase Similarity**.

# 4 CLIMAGEN - AUTOMATED BENCHMARK CREATION

In this section, we describe ClimaGen as shown in Figure 1, the framework used to create the ClimaQA dataset from Climate Textbooks. By leveraging RAG and prompt engineering, we systematically generate and refine questions of varying complexity levels. Domain expert annotators ensure quality, producing a semi-synthetic benchmark for the evaluation of AI systems in complex scientific inquiry. Additionally, we automate annotation by fine-tuning an LLM on human-labeled data, enabling the creation of a large-scale synthetic dataset for fine-tuning tasks. The techniques discussed here could be generalized to aid in the semi-automatic production of benchmarks for other scientific fields aswell.

## 4.1 TEXTBOOK DATASET

LLMs are typically pre-trained on extensive general internet data, which often contains noise and misinformation. This limitation is particularly significant in fields like climate science, where a precise understanding of specialized terminology and concepts is crucial. To evaluate LLMs' proficiency in climate science, we employed graduate-level climate science textbooks as a reliable source of specialized knowledge, providing accurate scientific information that better represents the technical terms and nuanced theories integral to this discipline. We collected 18 textbooks (Table 4) that broadly represent a mixture of graduate and expert literature on the physical climate with a particular focus on the role of aerosol in the climate system - one of the critical sources of uncertainty in climate projections. The content was extracted and preprocessed to ensure cleanliness and relevance, making it suitable for downstream applications such as benchmark creation, continuous pre-training, and RAG. A held-out set of 5 textbooks (Figure 6), carefully selected to represent varying levels of technical and qualitative depth across a broad range of key topics in climate science, was utilized for the benchmark creation process.

## 4.2 QA GENERATION FRAMEWORK

Our QA generation pipeline begins by selecting a random seed 2000-character context chunk from the collected textbook data stored in a vector database. Additional context chunks are retrieved based on cosine-similarity scores, ensuring relevant information from multiple sources is included. These chunks are then augmented and passed to the *generator* LLM for question-answer (QA) generation. Question generation principles, inspired by (Doughty et al., 2024), guide the prompt formulation, focusing on creating high-quality stems and distractors for MCQs, as well as refining questions by adding complexity as described in Appendix A.4.

We used GPT-3.5-turbo as the generator model in our experiments. The model generates base-level questions and evolved variants with increasing complexity, such as multi-step reasoning and hypothetical scenarios, to ensure a diverse and comprehensive question set as shown in Figure 2. However, approximately 25% of the multiple-choice questions were incorrectly answered by the same model even when the contexts were provided, often due to scientifically inaccurate question-answer pairs, indicating self-inconsistency and uncertainty in the generation process.

While QA pairs were generated for multiple-choice and freeform questions, plain scientific statements intended for cloze questions were also generated, from which the scientific term to be masked would be chosen during the annotation phase. After generation, the questions undergo a preliminary screening with both handcrafted and LLM-based (self-inconsistency) audits to filter out potentially invalid QA pairs. The refined set of QA pairs is then passed to the annotation phase for further validation.

## 4.3 DOMAIN EXPERT ANNOTATION

One key challenge with synthetic data is ensuring its distribution closely mirrors real-world data, as deviations can negatively impact downstream tasks like fine-tuning and evaluation. This issue arose when generating scientific questions with GPT-3.5-turbo, which sometimes produced inaccurate or imprecise data probably due to a limited understanding of domain-specific terminology.

To mitigate this, we developed an interactive web application that enables climate scientists to review and annotate the generated questions. For freeform and MCQs, the scientists validated the correctness of the content, while for cloze questions, they selected which scientific term to mask, ensuring alignment with scientific standards. They also identified common reasons for rejecting generated QA pairs during validation, providing valuable insights into improvement of the QA generation framework as discussed in Appendix A.5.1. By combining human expertise with AI, we curated 245 freeform, 161 MCQ, and 160 cloze questions, forming the *ClimaQA-Gold* dataset, reviewed and validated by domain experts.

### 4.4 AUTOMATED ANNOTATION

To fully automate the review and annotation process, we develop an *evaluator* model by fine-tuning an LLM (GPT-4o-mini) on expert-annotated data to validate and refine generated question-answer pairs. This removes the need for human intervention and enables scalable generation of high-quality scientific question-answer pairs, especially for data-intensive tasks like fine-tuning.

Building on (Zhang et al., 2023) and (Zhang et al., 2024), which demonstrate that uncertainty-based active sampling improves supervised fine-tuning with limited labeled data, we apply a similar approach. The evaluator model is fine-tuned as a classifier to label QA pairs as *valid* or *invalid* based on the given context as described in Appendix A.5.2. Uncertainty is measured by the classifier confidence scores, and samples with confidence above 0.85 are dropped with 50% probability to ensure learning from more representative examples. The evaluator models were fine-tuned separately for both MCQs and freeform questions

We observed that around 85% of multiple-choice questions (MCQs) and around 90% of freeform question-answers (QAs) were valid, indicating high-quality question generation. Experiments across different train-test splits show that the evaluator models enhance the quality of the generated MCQ question set by 10% and the Freeform question set by 5% as shown in Appendix A.5.2. Additionally, we fine-tune a separate model to mark scientific terms from given statements as shown in Appendix A.5.3, automating the cloze annotation process. Using this framework, we generated 1000 freeform QAs, 1000 MCQs, and 1000 cloze questions, collectively forming the *ClimaQA-Silver* dataset, produced synthetically at scale without manual intervention.

## 5 EXPERIMENTS

We aim to investigate the effectiveness of various adaptation techniques on this fine-grained scientific benchmark. Fine-tuning on raw text data within a target domain is a common approach, and we seek to evaluate its effectiveness for addressing deep scientific questions. In addition, we evaluate other techniques, such as in-context learning and retrieval augmentation.

### 5.1 EXPERIMENTAL SETUP

We evaluate different families of LLMs on our benchmark. We use TogetherAI for performing inference on open source models like gemma-27b (Team et al., 2024b), llama3-70b(Dubey et al., 2024), and mixtral-8x22b (Jiang et al., 2024). We also evaluate OpenAI's (Achiam et al., 2023) gpt-3.5-turbo and gpt-4o.

We evaluate each of these models in 3 settings - default, few-shot prompting (FS)(Brown, 2020), and Retrieval Augmented Generation (RAG) (Lewis et al., 2020). For the MCQs, the models were prompted to output a single letter representing the correct option, and the top-most token was chosen as the answer. For Freeform QA, the models were prompted to output concise answers with a maximum of 2 sentences. For Cloze QA, the models were prompted to output a single scientific word that best fits the blank with respect to the context around it.

We conduct further pre-training on graduate-level textbook data for both the LLaMA3.1-8B-Instruct and Mistral-7B-v0.3-Instruct (Jiang et al., 2023) models. This pre-training was based on 13 distinct graduate textbooks that were not part of the question-generation process. The objective was to enhance the model's climate knowledge without directly exposing it to the specific sources used for question generation, thereby reducing the risk of data contamination.

Table 3: Performance analysis of various state-of-the-art LLMs on MCQs and Cloze QA. While *source* represents the set of books used for QA generation, *held-out* represents the remaining set of books. Bold marks the max within a model's variants and green highlights the overall column max.

| Model | MCQ | | | | Cloze | |
|---|---|---|---|---|---|---|
| | Base | Reason | Hypo | Overall | EM | PS |
| gemma-27b | 81.75 | 72.22 | **82.98** | 79.18 | 49.38 | 0.87 |
| gemma-27b-fs | 80.95 | 77.78 | **82.98** | 80.41 | 52.50 | 0.88 |
| gemma-27b-rag-source | **90.48** | **80.56** | 78.72 | **85.31** | **56.88** | **0.90** |
| gemma-27b-rag-held-out | 79.37 | 76.39 | 78.72 | 78.37 | 45.62 | 0.85 |
| gpt-3.5-turbo | 74.34 | 69.91 | 74.47 | 73.06 | 43.12 | 0.81 |
| gpt-3.5-turbo-fs | 76.98 | 74.54 | 76.60 | 76.19 | 43.75 | 0.78 |
| gpt-3.5-turbo-rag-source | **80.42** | **80.09** | 77.30 | **79.73** | **68.75** | **0.92** |
| gpt-3.5-turbo-rag-held-out | 70.63 | 71.30 | 69.50 | 70.61 | 39.38 | 0.81 |
| gpt-4o | 86.77 | 86.11 | 82.27 | 85.71 | 53.12 | 0.88 |
| gpt-4o-fs | 87.83 | 87.50 | 80.85 | 86.39 | 56.25 | 0.89 |
| gpt-4o-rag-source | **95.77** | **91.67** | **86.52** | **92.79** | **71.88** | **0.94** |
| gpt-4o-rag-held-out | 82.80 | 80.56 | 81.56 | 81.90 | 50.62 | 0.88 |
| llama3-70b | 84.92 | 80.56 | 82.98 | 83.27 | 38.75 | 0.82 |
| llama3-70b-fs | 82.54 | 81.94 | 82.98 | 82.45 | 48.12 | 0.85 |
| llama3-70b-rag-source | **92.06** | **84.72** | **87.23** | **88.98** | **63.12** | **0.91** |
| llama3-70b-rag-held-out | 80.95 | 76.39 | 85.11 | 80.41 | 43.75 | 0.84 |
| mixtral-8x22b | 80.16 | 79.17 | **80.85** | 80.00 | 35.62 | 0.75 |
| mixtral-8x22b-fs | 80.95 | **81.94** | **80.85** | 81.22 | **45.00** | **0.83** |
| mixtral-8x22b-rag-source | **90.48** | 80.56 | 76.60 | **84.90** | **45.00** | 0.78 |
| mixtral-8x22b-rag-held-out | 80.16 | 73.61 | 74.47 | 77.14 | 28.12 | 0.65 |
| mistral-7b | 64.29 | 63.89 | 82.98 | 67.76 | 17.50 | 0.74 |
| mistral-7b-fs | 65.08 | 66.67 | 76.6 | 67.76 | 33.12 | 0.80 |
| mistral-7b-rag-source | **92.86** | **84.72** | **85.11** | **88.98** | **57.5** | **0.88** |
| mistral-7b-rag-held-out | 66.67 | 62.5 | 74.47 | 66.94 | 22.50 | 0.76 |
| mistral-7b-cp-held-out | 72.22 | 62.5 | 74.47 | 69.8 | 21.25 | 0.76 |
| mistral-7b-ft-silver | 73.81 | 75.00 | 80.85 | 75.51 | 41.88 | 0.83 |
| llama3.1-8b | 76.98 | 62.50 | 65.96 | 70.61 | 26.25 | 0.77 |
| llama3.1-8b-fs | 76.19 | 72.22 | 76.60 | 75.10 | 38.75 | 0.82 |
| llama3.1-8b-rag-source | **94.44** | **83.33** | **89.36** | **90.2** | **72.50** | **0.92** |
| llama3.1-8b-rag-held-out | 76.19 | 66.67 | 76.6 | 73.47 | 36.25 | 0.81 |
| llama3.1-8b-cp-held-out | 77.78 | 75.00 | 72.34 | 75.92 | 30.00 | 0.77 |
| llama3.1-8b-ft-silver | 74.6 | 70.83 | 72.34 | 73.06 | 51.25 | 0.85 |
| gemini-1.5-flash | 82.54 | 73.61 | **87.23** | 80.82 | 50.62 | 0.88 |
| gemini-1.5-flash-long-cxt-source | **88.1** | **75.00** | 80.85 | **82.86** | **51.88** | **0.89** |
| gemini-1.5-flash-long-cxt-held-out | 70.63 | 70.83 | 78.72 | 72.24 | 46.88 | 0.87 |

Additionally, we fine-tune LLaMA3.1-8B-Instruct and Mistral-7B-v0.3-Instruct on the ClimaQA-Silver dataset, which contains all three forms of MCQ, Freeform, and Cloze, in different complexity levels. We then evaluate the impact of this task-specific fine-tuning by assessing the models' performance on the ClimaQA-Gold dataset. The details of the continued pre-training and fine-tuning procedure are explained in Appendix A.3. We also leverage Gemini 1.5 (Team et al., 2024a), with a context window of up to 1 million tokens, to pass an entire textbook in context and answer questions based on that. Finally, we evaluate models on the ClimaQA-Silver dataset and analyze its potential differences from ClimaQA-Gold in Appendix A.2.2.

## 5.2 RESULTS

We report the performance of various models across different QA forms and complexities. Table 3 shows the results for the MCQ and Cloze form questions , and free-form results are demonstrated in Appendix Table 5. We observe that most models struggle with reasoning MCQs compared to base and hypothetical questions. While some models perform poorly on reasoning and hypothetical

MCQs, they tend to generate stronger responses for the same type of Freeform Questions, indicating improved performance when reasoning is emphasized over factual recall similar to observations of Chain of Thought experiments (Wei et al., 2022), as shown in Figure 5.

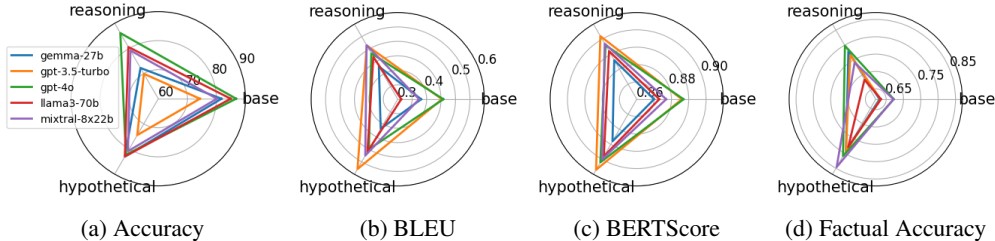

(a) Accuracy    (b) BLEU    (c) BERTScore    (d) Factual Accuracy

Figure 5: Analysis of various LLMs under default setting on different tasks and different complexities. The first figure shows accuracy of models in the MCQ task while the others show different metrics under the Freeform task

We examine the impact of providing relevant context through RAG when answering questions. We explore two retrieval scenarios: one where the model retrieves from 13 books that were not used to generate the questions (rag-held-out), and another where it retrieves from the 5 books that the questions were derived from (rag-source). Retrieval from source textbooks consistently enhances performance across all tasks. For Gemini 1.5, we identify the book corresponding to the most relevant retrieved chunk under both the source and held-out settings, and include it as in-context information to answer the question. Additional context from source books yields a slight improvement over the no-context baseline. However, in both RAG and long-context scenarios, incorporating held-out books usually reduces performance, likely due to irrelevant or distracting content.

Continued pertaining (cp-held-out) on the graduate textbooks leads to improved performance in both MCQs and Cloze question-answering tasks. Additionally, fine-tuning (ft-silver) on the ClimaQA-Silver dataset further enhances performance, often producing the best results after RAG on the source textbooks (rag-source) in most scenarios. Furthermore, Few-shot prompting yields marginal improvements in most cases.

Finally, We observe that the BLEU and BERTScore metrics are slightly biased towards the model that was used for QA-generation (gpt-3.5-turbo) while this is not seen in the proposed Factual Accuracy metric 5. Overall, GPT-4o dominates across tasks, demonstrating superior performance compared to other models in this evaluation set.

## 6    CONCLUSION

The *ClimaQA* benchmark offers a comprehensive framework for evaluating language models in climate question-answering, addressing critical aspects such as reasoning, factual accuracy, and understanding of scientific terminology. By incorporating freeform, multiple-choice, and cloze task forms with different levels of complexity, the benchmark rigorously tests models across different dimensions of scientific inquiry. Furthermore, the use of advanced metrics, such as factual accuracy for freeform tasks and phrase similarity for cloze tasks, can provide a more nuanced assessment of model performance.

The automated benchmark generation framework (ClimaGen) integrates domain-specific textbooks and natural language understanding of LLMs along with human expertise to produce high-quality QA data at scale. However, the benchmark's reliance on only five textbooks limits the diversity of contexts, and the small size of the annotated dataset constrains the effectiveness of automated annotation. Addressing these limitations with a broader corpus and expanded annotation data will improve future benchmarks.

While models like GPT-4o performed well on reasoning-based tasks, the overall performance of models highlights the ongoing challenge of achieving consistent scientific accuracy. In conclusion, *ClimaQA* sets a new standard for evaluating scientific question-answering models, providing a foundation for future advancements in AI-driven climate research.

## REPRODUCIBILITY STATEMENT

The ClimaQA dataset, both gold and silver, is publicly available at Hugging Face[1]. While we cannot release the scraped textbook data due to copyright restrictions, the references to all textbooks used are provided in the appendix A.1, allowing for reconstruction of this dataset. Our complete codebase, including data generation pipeline, web-UI and model evaluation scripts, is available in our GitHub repository[2]. The ClimaGen framework is fully reproducible; however, training the evaluator models requires domain expert inputs, which may introduce variability. Furthermore, reproduction of all parts of our framework requires appropriate API keys for external services.

## ACKNOWLEDGEMENT

This work was supported in part by the U.S. Army Research Office under Army-ECASE award W911NF-07-R-0003-03, the U.S. Department Of Energy, Office of Science, IARPA HAYSTAC Program, and NSF Grants #2205093, #2146343, #2134274, CDC-RFA-FT-23-0069, DARPA AIE FoundSci and DARPA YFA. We thank Sophie Wynn, Varan Madan, and David Vishny for providing expert validation of the *ClimaQA* dataset.

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

# A APPENDIX

## A.1 TEXTBOOK DATASET

Table 4: List of climate textbooks used in this work along with the number of pages and the number of context chunks. The extracted data was preprocessed to omit figures and tables. The content of each page was then split into overlapping context chunks of around 2000 characters.

| Textbook | Pages | Chunks |
| --- | --- | --- |
| Aerosol Measurement (Baron & Willeke, 2001) | 937 | 1813 |
| Aerosols and Climate (Carslaw, 2022) | 854 | 1766 |
| Airborne CCN Measurements (Trembath, 2013) | 268 | 333 |
| An Introduction to Clouds (Lohmann et al., 2016) | 377 | 606 |
| Atmospheric Chemistry and Physics (Seinfeld & Pandis, 2016) | 1127 | 1789 |
| Atmospheric Science (Wallace & Hobbs, 2006) | 495 | 1085 |
| Calculus of Variations (Gelfand & Fomin, 2012) | 239 | 266 |
| Clouds in the Perturbed Climate System (Heintzenberg & Charlson, 2009) | 598 | 1060 |
| Eloquent Science (Schultz, 2013) | 420 | 725 |
| Filtering Complex Turbulent Systems (Majda & Harlim, 2012) | 358 | 475 |
| Fundamental of Atmospheric Modeling (Jacobson, 2005) | 826 | 1204 |
| Geostatistics for Environmental Scientists (Webster & Oliver, 2007) | 327 | 457 |
| Global Physical Climatology (Hartmann, 2015) | 481 | 715 |
| Principles Of Planetary Climate (Pierrehumbert, 2010) | 468 | 1096 |
| Forests and Climate Change (De Wasseige et al., 2015) | 119 | 250 |
| Simulating Nature (Petersen, 2012) | 205 | 386 |
| Statistical Methods in the Atmospheric Sciences (Wilks, 2019) | 806 | 1597 |
| Stochastic Climate Models (Imkeller & von Storch, 2012) | 411 | 594 |

To assess the proficiency of LLMs in climate science, we utilized graduate-level climate science textbooks as a reliable source of specialized knowledge. These textbooks were selected for their accurate and comprehensive representation of the technical terminology and nuanced theories integral to the field. The collection was curated from the virtual bookshelf of a professor in atmospheric physics and broadly represents a mixture of graduate and expert textbooks on the physical climate, with a particular focus on the role of aerosol in the climate system (which provides one of the key uncertainties in climate projections). The complete list of textbooks is provided in Table 4.

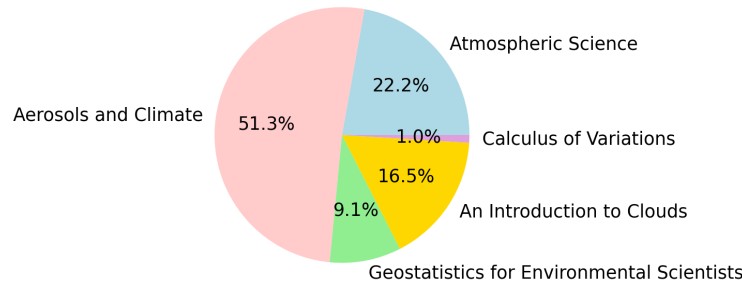

Figure 6: The distribution of ClimaQA-Gold questions over the corresponding source.

The five textbooks selected for question generation were carefully chosen to reflect different levels of breadth and depth (both technically and more qualitatively) across a range of important topics in climate science. These include *Atmospheric Science*, a comprehensive graduate-level textbook on atmospheric science; *Aerosols and Climate*, a brand-new and more detailed textbook on the role of aerosol in the climate system; *An Introduction to Clouds*, another relatively new textbook providing the latest research on the important role of clouds in the climate system; *Geostatistics for*

*Environmental Scientists*, a more technical graduate-level textbook on geostatistics for environmental sciences; as well as *Calculus of Variations*, a classic calculus textbook to test the extent of the technical ability of the models. The exact distribution of questions from these textbooks is shown in Figure 6. The remaining textbooks were used in the rag-held-out experiments to test the models without directly exposing them to the sources of the benchmark.

## A.2 MORE EXPERIMENTS

### A.2.1 FREEFORM EVALUATION RESULTS

Table 5: Performance analysis of various state-of-the-art LLMs on Freeform QA. Bold marks the max within a model's variants and green highlights the overall column max.

| Model | BLEU | | | | BERTScore | | | | Factual Acccuracy | | | |
|---|---|---|---|---|---|---|---|---|---|---|---|---|
| | B | R | H | O | B | R | H | O | B | R | H | O |
| gemma-27b | 0.381 | 0.431 | 0.418 | 0.410 | 0.870 | 0.886 | 0.888 | 0.881 | 0.63 | 0.78 | 0.79 | 0.73 |
| gemma-27b-fs | 0.430 | 0.400 | 0.361 | 0.397 | 0.884 | 0.895 | 0.896 | 0.892 | 0.65 | 0.77 | 0.82 | 0.75 |
| gemma-27b-rag-source | **0.462** | **0.573** | **0.552** | **0.529** | **0.902** | **0.911** | **0.906** | **0.906** | **0.84** | **0.85** | **0.85** | **0.85** |
| gemma-27b-rag-held-out | 0.349 | 0.496 | 0.524 | 0.456 | 0.867 | 0.889 | 0.896 | 0.885 | 0.58 | 0.73 | 0.77 | 0.69 |
| gpt-3.5-turbo | 0.453 | 0.515 | **0.579** | 0.516 | 0.886 | 0.902 | 0.907 | 0.899 | 0.63 | 0.77 | 0.79 | 0.73 |
| gpt-3.5-turbo-fs | 0.514 | 0.451 | 0.465 | 0.477 | 0.894 | 0.908 | 0.915 | 0.906 | 0.61 | 0.77 | 0.86 | 0.75 |
| gpt-3.5-turbo-rag-source | **0.528** | **0.601** | 0.561 | **0.563** | **0.908** | **0.919** | **0.921** | **0.916** | **0.85** | **0.86** | **0.87** | **0.86** |
| gpt-3.5-turbo-rag-held-out | 0.425 | 0.477 | 0.543 | 0.482 | 0.877 | 0.889 | 0.904 | 0.891 | 0.46 | 0.64 | 0.81 | 0.64 |
| gpt-4o | 0.458 | 0.483 | 0.500 | 0.480 | 0.887 | 0.896 | 0.902 | 0.895 | 0.67 | 0.80 | 0.81 | 0.76 |
| gpt-4o-fs | 0.502 | 0.430 | 0.411 | 0.448 | 0.898 | 0.906 | 0.914 | 0.906 | 0.66 | 0.82 | 0.84 | 077 |
| gpt-4o-rag-source | **0.539** | **0.626** | **0.620** | **0.595** | **0.916** | **0.917** | **0.916** | **0.916** | **0.81** | **0.83** | 0.85 | **0.83** |
| gpt-4o-rag-held-out | 0.425 | 0.520 | 0.561 | 0.502 | 0.884 | 0.892 | 0.903 | 0.893 | 0.54 | 0.71 | **0.86** | 0.70 |
| llama3-70b | 0.312 | 0.469 | 0.510 | 0.430 | 0.873 | 0.892 | 0.898 | 0.888 | 0.65 | 0.66 | **0.85** | 0.72 |
| llama3-70b-fs | **0.456** | 0.557 | 0.585 | 0.532 | 0.889 | 0.903 | 0.909 | 0.900 | 0.64 | 0.68 | 0.78 | 0.70 |
| llama3-70b-rag-source | 0.441 | **0.574** | **0.604** | **0.540** | **0.909** | **0.916** | **0.917** | **0.914** | **0.82** | **0.81** | 0.84 | **0.82** |
| llama3-70b-rag-held-out | 0.341 | 0.475 | 0.519 | 0.445 | 0.872 | 0.890 | 0.902 | 0.888 | 0.45 | 0.62 | 0.79 | 0.62 |
| mixtral-8x22b | 0.374 | 0.516 | 0.525 | 0.471 | 0.877 | 0.897 | 0.900 | 0.891 | 0.67 | 0.74 | 0.85 | 0.75 |
| mixtral-8x22b-fs | **0.495** | 0.497 | 0.486 | 0.493 | 0.892 | 0.906 | **0.913** | 0.904 | 0.68 | 0.76 | 0.83 | 0.76 |
| mixtral-8x22b-rag-source | 0.458 | **0.598** | **0.610** | **0.555** | **0.905** | **0.916** | 0.912 | **0.911** | **0.78** | **0.82** | **0.86** | **0.82** |
| mixtral-8x22b-rag-held-out | 0.341 | 0.488 | 0.546 | 0.459 | 0.870 | 0.890 | 0.898 | 0.886 | 0.50 | 0.62 | 0.77 | 0.63 |
| llama3.1-8b | 0.387 | 0.467 | 0.505 | 0.453 | 0.872 | 0.889 | 0.899 | 0.887 | 0.59 | 0.65 | 0.75 | 0.66 |
| llama3.1-8b-fs | 0.399 | 0.484 | 0.532 | 0.472 | 0.877 | 0.894 | 0.905 | 0.892 | 0.59 | 0.68 | 0.75 | 0.67 |
| llama3.1-8b-rag-source | **0.509** | **0.550** | **0.534** | **0.531** | **0.905** | **0.920** | **0.911** | **0.912** | **0.77** | **0.80** | **0.83** | **0.80** |
| llama3.1-8b-rag-held-out | 0.392 | 0.469 | 0.507 | 0.456 | 0.873 | 0.893 | 0.901 | 0.889 | 0.52 | 0.65 | 0.72 | 0.63 |
| llama3.1-8b-cp-held-out | 0.420 | 0.455 | 0.496 | 0.457 | 0.876 | 0.892 | 0.902 | 0.890 | 0.53 | 0.70 | 0.71 | 0.65 |
| llama3.1-8b-ft-silver | 0.436 | 0.469 | 0.533 | 0.480 | 0.882 | 0.896 | 0.904 | 0.894 | 0.53 | 0.59 | 0.77 | 0.63 |
| mistral-7b | 0.385 | 0.370 | 0.412 | 0.390 | 0.869 | 0.886 | 0.892 | 0.882 | 0.58 | 0.64 | 0.74 | 0.65 |
| mistral-7b-fs | 0.432 | 0.409 | 0.438 | 0.427 | 0.878 | 0.901 | **0.908** | 0.896 | 0.51 | 0.66 | 0.80 | 0.66 |
| mistral-7b-rag-source | **0.541** | **0.455** | **0.448** | **0.482** | **0.898** | **0.912** | 0.907 | **0.906** | **0.77** | **0.82** | **0.84** | **0.81** |
| mistral-7b-rag-held-out | 0.413 | 0.356 | 0.399 | 0.390 | 0.868 | 0.884 | 0.896 | 0.883 | 0.54 | 0.60 | 0.73 | 0.62 |
| mistral-7b-cp-held-out | 0.295 | 0.176 | 0.189 | 0.221 | 0.880 | 0.890 | 0.899 | 0.890 | 0.52 | 0.70 | 0.70 | 0.64 |
| mistral-7b-ft-silver | 0.520 | 0.386 | 0.434 | 0.447 | 0.892 | 0.900 | 0.906 | 0.899 | 0.40 | 0.61 | 0.82 | 0.61 |

### A.2.2 CLIMAQA-SILVER EXPERIMENTS

In this section, we assess the models in their default settings on a subset of the ClimaQA-Silver dataset, consisting of 200 questions for each task type. The column-wise trends observed in the results are largely consistent with those from the ClimaQA-Gold dataset. However, a notable difference lies in the relative difficulty of the complexities in the MCQ task, even though the overall scores do not vary significantly. This discrepancy highlights the intricate nature of the MCQ generation process. Additionally, this variation may also be attributed to the relatively small number of questions in each column. Future work should focus on scaling up the expert validation process to enhance the quality of the automated annotation pipeline, thereby addressing these challenges and improving overall dataset reliability.

Table 6: Performance analysis of various LLMs on MCQs and Cloze QA in ClimaQA-Silver

| Model | MCQ | | | | Cloze | |
|---|---|---|---|---|---|---|
| | Base | Reason | Hypo | Overall | EM | PS |
| gemma-27b | 78.00 | **84.91** | 76.6 | 79.5 | 50.00 | 0.85 |
| gpt-3.5-turbo | 76.00 | 75.47 | 72.34 | 75.0 | 38.00 | 0.78 |
| gpt-4o | **88.00** | **84.91** | 78.72 | **85.00** | **60.50** | **0.88** |
| llama3-70b | 85.00 | 79.25 | 65.96 | 79.00 | 44.00 | 0.83 |
| mixtral-8x22b | 80.00 | 81.13 | **82.98** | 81.00 | 33.00 | 0.67 |

Table 7: Performance analysis of various LLMs on Freeform QA in ClimaQA-Silver

| Model | BLEU | | | | BERTScore | | | | Factual Acccuracy | | | |
|---|---|---|---|---|---|---|---|---|---|---|---|---|
| | B | R | H | O | B | R | H | O | B | R | H | O |
| gemma-27b | 0.392 | 0.441 | 0.365 | 0.398 | 0.870 | 0.880 | 0.886 | 0.876 | 0.78 | 0.86 | 0.85 | 0.81 |
| gpt-3.5-turbo | **0.467** | **0.523** | 0.545 | **0.500** | **0.885** | **0.892** | **0.907** | **0.892** | 0.71 | 0.79 | 0.84 | 0.76 |
| gpt-4o | 0.440 | 0.493 | 0.491 | 0.465 | 0.880 | 0.887 | 0.902 | 0.887 | **0.80** | **0.88** | **0.86** | **0.84** |
| llama3-70b | 0.335 | 0.474 | **0.569** | 0.425 | 0.874 | 0.888 | 0.904 | 0.885 | 0.77 | 0.82 | 0.75 | 0.78 |
| mixtral-8x22b | 0.394 | 0.516 | 0.485 | 0.447 | 0.877 | 0.890 | 0.904 | 0.887 | **0.80** | **0.88** | 0.83 | 0.82 |

### A.2.3 CASE STUDY

In this section, we present sample questions from the ClimaQA-Gold dataset along with responses from various LLMs and their corresponding evaluation scores.

Freeform QA

In this example, the reference answer discusses increased global emissions in the absence of air quality measures. While the outputs of most LLMs align with this reference, the response from *gemma-27b* suggests a decline in global emissions. Note that both the Factual Accuracy and BLEU metrics penalize this response, whereas the BERTScore does not.

> **Quesition:** How might the global emission estimates have been different if air quality measures were not introduced mainly targeted at large, coal-burning installations before 2000?
>
> **Reference Answer:** Without the introduction of air quality measures mainly targeted at large, coal-burning installations before 2000, the global emission estimates, particularly for SO2, may not have shown a decrease before 2000. The emissions could have continued to rise or remained at higher levels due to the ongoing industrial growth and lack of regulations to curb pollution from such sources.

**gpt-3.5-turbo:** Global emission estimates would likely have been higher if air quality measures targeting large, coal-burning installations were not introduced before 2000, as these measures have helped to reduce emissions of pollutants such as sulfur dioxide and particulate matter.

*Bleu:* 0.485, *Bert:* 0.901, *Factual Accuracy:* 0.988

**gpt-4o:** Without air quality measures targeting large, coal-burning installations before 2000, global emissions, particularly sulfur dioxide and particulate matter, would likely have been significantly higher, exacerbating air pollution and related health issues. This could have also contributed to more severe acid rain and potentially altered climate patterns due to increased atmospheric aerosols.

*Bleu:* 0.589, *Bert:* 0.898, *Factual Accuracy:* 0.982

**llama3-70b:** If air quality measures were not introduced before 2000, global emission estimates would likely be higher, as coal-burning installations would have continued to emit large amounts of pollutants, including particulate matter, sulfur dioxide, and nitrogen oxides, contributing to increased greenhouse gas emissions. This could have resulted in an additional 1-2 gigatons of CO2 equivalent emissions per year, exacerbating climate change impacts.

*Bleu:* 0.542, *Bert:* 0.888, *Factual Accuracy:* 0.968

**mixtral-8x22b:** If air quality measures hadn't been introduced for large, coal-burning installations before 2000, global emission estimates would likely have been significantly higher due to unregulated emissions from these sources. This could have potentially accelerated global warming and worsened air quality on a global scale.

*Bleu:* 0.546, *Bert:* 0.910, *Factual Accuracy:* 0.988

**gemma-27b:** Global emission estimates would likely be significantly higher without those air quality measures. Reduced emissions from coal-burning power plants would have contributed less to the overall decline in global emissions observed since 2000.

*Bleu:* 0.364, *Bert:* 0.885, *Factual Accuracy:* 0.741

In the following example, the reference answer indicates that the distance from the sun influences surface pressure, with higher pressures during summer. Responses from *gpt-3.5-turbo* and *mixtral-8x22b* contradict this statement but still achieve high BLEU and BERTScore values. The Factual Accuracy metric, however, correctly identifies and penalizes this inconsistency. Notably, *gpt-4o* provides a correct answer yet receives a low BLEU score, as BLEU is not robust against paraphrasing.

**Question:** How does the distance from the sun affect the surface pressure on Mars?

**Reference Answer:** The surface pressure on Mars varies seasonally from 7.0 hPa during the austral winter, when Mars is farthest from the sun, to 9.0 hPa during the austral summer.

**gpt-3.5-turbo:** The distance from the sun does not directly affect the surface pressure on Mars; instead, the surface pressure on Mars is primarily influenced by the planet's atmospheric composition and temperature.

*Bleu:* 0.388, *Bert:* 0.880, *Factual Accuracy:* 0.175

**gpt-4o:** The distance from the Sun affects Mars' temperature, which in turn influences the sublimation of its polar ice caps, impacting atmospheric pressure. During warmer periods, more CO2 sublimates, increasing surface pressure, while colder periods result in lower pressure.

*Bleu:* 0.253, *Bert:* 0.865, *Factual Accuracy:* 0.980

**llama3-70b:** The distance from the Sun affects the surface pressure on Mars by controlling the amount of atmospheric gas that freezes out or escapes to space, with more gas freezing out at greater distances. As a result, Mars' surface pressure is about 1% of Earth's, due to its average distance from the Sun being about 1.5 times greater than Earth's.

*Bleu:* 0.262, *Bert:* 0.861, *Factual Accuracy:* 0.896

**mixtral-8x22b:** The distance from the sun does not significantly affect the surface pressure on Mars. Instead, Mars' thin atmosphere and low gravity are the primary factors influencing its surface pressure.

*Bleu:* 0.403, *Bert:* 0.874, *Factual Accuracy:* 0.051

**gemma-27b:** Mars is farther from the sun than Earth, receiving less solar radiation. This weaker solar heating results in a much lower surface pressure on Mars compared to Earth.

*Bleu:* 0.458, *Bert:* 0.870, *Factual Accuracy:* 0.905

Interestingly, *gemma-27b* provides an irrelevant response that nevertheless receives high scores. While the response is factually accurate, it fails to address the question, illustrating limitations in existing evaluation metrics. This highlights the need for more fine-grained evaluation methodologies, which future research should aim to develop.

Cloze QA

In the following examples, the models generate answers that differ from the reference answer but remain correct within the given context. This is effectively captured by the phrase similarity metric, which assigns higher scores to responses that are contextually aligned with the reference answer.

**Question:** Aqueous-phase oxidation of $\langle mask \rangle$ to form sulfuric acid in cloud droplets is a crucial process that contributes significantly to global sulfate mass and has long been recognized as important for aerosols and climate.

**Reference Answer:** SO2

**gpt-3.5-turbo:** Sulfur, *Phrase Similarity:* 0.819

**gpt-4o:** sulfur dioxide, *Phrase Similarity:* 0.885

**llama3-70b:** Sulfur, *Phrase Similarity:* 0.819

**mixtral-8x22b:** sulfur dioxide, *Phrase Similarity:* 0.885

**gemma-27b:** sulfur dioxide, *Phrase Similarity:* 0.885

**Question:** The rate of change in the air parcel's moist static energy due to $\langle mask \rangle$ is determined by the difference between the moist static energies of the environment and the cloud air parcel.

**Reference Answer:** entrainment

**gpt-3.5-turbo:** evaporation, *Phrase Similarity:* 0.681

**gpt-4o:** entrainment, *Phrase Similarity:* 1.000

**llama3-70b:** entrainment, *Phrase Similarity:* 1.000

**mixtral-8x22b:** mixing, *Phrase Similarity:* 0.726

**gemma-27b:** entrainment, *Phrase Similarity:* 1.000

## A.3 TRAINING DETAILS

We used Llama3.1-8B and Mistral-7B-v0.3 as our base models and performed continued pre-training and fine-tuning on them. We utilized the Low-Rank Adaptation (LoRA) (Hu et al., 2021) technique for efficient continued pre-training and fine-tuning.

### A.3.1 CONTINUED PRE-TRAINING ON GRADUATE TEXTBOOK DATA

We used 13 graduate textbooks as our training set, while the 5 textbooks used for question generation served as the validation set. A hyperparameter search was performed, guided by the cross-entropy loss on the validation set of 5 textbooks. The final hyperparameters used for continued pre-training are detailed in Table 8.

| Model | LoRA Rank | LoRA Alpha | Epoch Count | Learning Rate |
|---|---|---|---|---|
| Mistral-7b-v0.3 | 64 | 16 | 1 | 5e-5 |
| Llama-3.1-8b | 16 | 16 | 2 | 2e-5 |

Table 8: Parameters used for graduate textbook continued pre-training

### A.3.2 FINE-TUNING ON CLIMAQA-SILVER

We used the ClimaQA-Silver dataset to fine-tune Llama3.1-8B and Mistral-7B-v0.3 on questions with the various question forms and complexities seen in the ClimaQA-Gold to see how this task-specific fine-tuning can affect the performance.

For this purpose, we used the hyperparameters, shown in Table 9.

| Model | LoRA Rank | LoRA Alpha | Epoch Count | Learning Rate |
|---|---|---|---|---|
| Mistral-7b-v0.3 | 16 | 16 | 3 | 5e-5 |
| Llama-3.1-8b | 16 | 16 | 3 | 5e-5 |

Table 9: Parameters used for question finetuning

## A.4 CLIMAGEN PROMPTS

### A.4.1 QA GENERATION

The following prompt was used to generate a multiple-choice question-answer pair:

> You are a question paper setter creating multiple choice questions (MCQs) from a graduate-level climate science textbook.
>
> MCQ Components:
>
> 1. Stem: The main question, scenario, or statement requiring completion. It should clearly assess the intended knowledge.
>
> 2. Correct Answer: The indisputable correct response to the stem.
>
> 3. Distractors: Three incorrect but plausible answers. They should be:
>
> - Related to the stem and correct answer.
>
> - Positively phrased and true statements that don't answer the stem.
>
> - Plausible but incorrect, without giving clues to the correct answer.
>
> - Unique, each reflecting different misconceptions if possible.
>
> MCQ Guidelines:

1. Questions should be clear, concise, and free from unnecessary complexity or ambiguity.

2. Avoid overly long sentences and use consistent phrasing for repeated items.

3. Ensure questions are self-contained and provide all necessary context.

4. Do not include phrases like "According to the provided context..."

5. Do not make any references to the given context in the question

6. Ensure that distractors do not overlap by reflecting different misconceptions on the topic.

7. Minimize clues that make the correct answer obvious.

8. Use "None of the Above" or "All of the Above" sparingly.

9. Each MCQ must have exactly four answer choices (one correct, three distractors).

10. Questions should not rely on external figures or tables.

The user will provide one main context and some retrieved contexts separated by '————————' as the input.

Use details from retrieved context only if they are relevant to your question.

You must output a single JSON object in the following format:

{
question: $\langle question \rangle$,

options: {
a: $\langle option1 \rangle$,
b: $\langle option2 \rangle$,
c: $\langle option3 \rangle$,
d: $\langle option4 \rangle$
},

correct option: c
}

Here c is the correct answer. Replace it with the actual correct answer.

Make sure you return a valid JSON object.

The following prompt was used to generate a freeform question-answer pair:

You are a question paper setter creating freeform questions (MCQs) from a graduate-level climate science textbook. Your question must be related to a provided context.

Please respect the following rules to generate the question:

- The answer to the question should be found inside the provided context

- The question must be self-contained

- Do not include phrases such as "According to the provided context.."

The user will provide one main context and some retrieved contexts separated by '————————' as the input.

Use details from retrieved context only if they are relevant to your question.

You must output a single JSON objectin the following format:

{

question: $\langle question \rangle$,

answer: $\langle answer \rangle$
}

Make sure you return a valid JSON object.

---

The following prompt was used to generate a scientific statement for cloze question-answer generation:

You are a scientific annotator. Given a scientific context from a climate textbook, generate a scientific statement based on the facts presented in the context.

Please respect the following rules to generate the statement:

- Generate only a single sentence

- No external knowledge should be used or refered in generating the statement

- Do not use phrases like 'based on the provided context.'

The user will provide one main context and some retrieved contexts seperated by '——————————' as the input.

Use details from retrieved context only if they are relevant to your question.

You must output a single JSON objectin the following format:

{

statement: $\langle statement \rangle$,

}

Make sure you return a valid JSON object.

### A.4.2 COMPLEXITY ADDITION

The following prompt was used to add *reasoning* complexity to the base freeform question-answer pair.

Given a question-answer pair generated from the given context, Modify the question-answer pair to incorporate multi-step reasoning.

Please respect the following rules to generate the question:

- Answering the new question should encourage applying knowledge from 'Context' to deduce outcomes.

- The new question must be fully answerable from 'Context'.

- No external knowledge should be required to answer the new question

- The question should not be dependent on external things such as figures or tables

- Do not use phrases like 'based on the provided context.'

The user will provide the original question, context, and some retrieved contexts separated by '————————————' as the input.

Use details from retrieved context only if they are relevant to your question.

You must output a single JSON objectin the following format:

```
{
question: ⟨question⟩,

answer: ⟨answer⟩
}

Make sure you return a valid JSON object.
```

The following prompt was used to add *hypothetical scenario* complexity to the base freeform question-answer pair.

Given a question-answer pair generated from the given context, Modify the question-answer pair to incorporate a hypothetical or speculative scenario.

Please respect the following rules to generate the question:

- Answering the new question should encourage applying knowledge from 'Context' to deduce outcomes.

- The new question must be fully answerable from 'Context'.

- No external knowledge should be required to answer the new question

- The question should not be dependent on external things such as figures or tables

- Do not use phrases like 'based on the provided context.'

The user will provide the original question, context, and some retrieved contexts separated by '————————' as the input.

Use details from retrieved context only if they are relevant to your question.

You must output a single JSON object in the following format:

{
question: ⟨question⟩,

answer: ⟨answer⟩
}

Make sure you return a valid JSON object.

The same prompts with modified output format were used to add complexities to the MCQs as well.

## A.5    ANNOTATION

### A.5.1    EXPERT VALIDATION

During the validation process of the ClimaQA-Gold dataset for both multiple-choice and freeform questions, experts were asked to select predefined reasons for rejecting a generated QA pair. They also had the option to provide a custom reason for rejection. The figure below illustrates the various types of errors made by the ClimaGen pipeline during the generation phase. While most rejection reasons highlight limitations of the generator LLM, the category of *bad context* points to instances where the seed context used for QA generation was inherently flawed or lacked meaningful information. Addressing this issue through improved preprocessing techniques is a key area for future work.

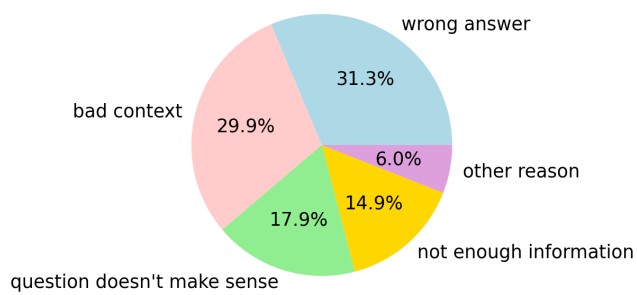

Figure 7: The distribution of different reasons selected for scientific invalidity of generated QA pairs by domain experts.

### A.5.2 AUTOMATED ANNOTATION - MCQ AND FREEFORM

We used *gpt-4o-mini* as the base model for the Evaluator Model. It was fine-tuned for the classification task with the prompt below. The same prompt was used for both MCQ and freeform validation and 2 different models were created by fine-tuning the base model with respective datasets.

> You are a climate {*question-type*} question-answer validator that marks a given question-answer pair as valid or invalid based on scientific accuracy with respect to the given context. You will be provided the following as the input:
>
> **Question**: $\langle question \rangle$
> **Answer**: $\langle answer \rangle$
> **Context**: $\langle context \rangle$
>
> Respond with just one word - VALID if the qa pair is scientifically accurate and INVALID otherwise

| Dataset | Valid Count | Invalid Count | Total | Epochs |
|---------|-------------|---------------|-------|--------|
| MCQ | 245 | 47 | 292 | 3 |
| Freeform | 161 | 20 | 181 | 3 |

Table 10: Details of expert-validated dataset and fine-tuning

Since negative samples are dropped in large-scale data generation, precision is used over accuracy as the key metric to measure the classifier's impact, as it represents the fraction of final output data that is valid.

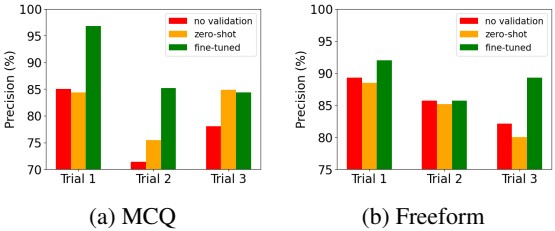

(a) MCQ  (b) Freeform

Figure 8: Precision across various train-test splits of labeled data under different conditions: *no validation* (all data classified as positive), *zero-shot validation*, and *fine-tuned validation*. The test sets consisted of around 40 examples in each case.

### A.5.3 AUTOMATED ANNOTATION - CLOZE

To automate cloze annotation, we fine-tuned *gpt-4o-mini* with the following prompt to pick the scientific term to be masked. The dataset consisted of a total of 160 expert-labeled examples.

You are a climate cloze generator that marks a scientific term from the given scientific statement to be masked for cloze question-answering The scientific term has to be a single word from the given statement that has a significant impact if removed You will be provided the following as the input:

**Statement**: $\langle statement \rangle$

Respond with just one word

