# OpenReview forum: "ClimaQA: An Automated Evaluation Framework for Climate Question Answering Models"
_ICLR.cc/2025/Conference — ICLR 2025 Poster_

### Official Review · Reviewer_mkPm · 2024-10-31

**Soundness:** 2
**Presentation:** 2
**Contribution:** 2
**Rating:** 6
**Confidence:** 5

**Summary:**

This paper designs an automated pipeline for constructing a ClimaQA benchmark, then presents two datasets, ClimaQA-Gold and ClimaQA-Silver, and finally conducts an experimental evaluation of existing LLMs on this benchmark.

**Strengths:**

- S1: This paper develops ClimaGen, an automated climate QA generator.
- S2: This paper proposes two datasets and three question-answering tasks.
- S3: Based on the ClimaQA benchmark, this paper presents some experiments to evaluate existing llms.

**Weaknesses:**

- W1: One of the core innovations of the paper is the tool ClimaGen that automates the construction of ClimaQA, which involves the QA principle and prompt assembler. From line345 it appears to be a direct reference to other articles, which the authors do not describe in detail. However, this part directly affects the quality of the input prompt, which determines the quality of the generated QA pairs.
- W2: This paper proposes a benchmark to assess the quality and scientific validity of the output of climate base models, but there is a lack of experimental validation that the QA pairs in the benchmark are scientifically meaningful, i.e., how to prove that the constructed dataset is able to help future research on climate foundation models, so that the output is of higher quality and scientific validity.
- W3: Lack of detailed information about the data source. The authors emphasize the data sources as 18 graduate-level textbooks, but give no detailed information about the data sources to prove it.
- W4: The experimental section lacks a case study. There is no visual comparison between the actual performance of the existing LLMs on these tasks and the difference between the groundtruth.
- W5: Writing flaws: lack of citations for some important ideas, such as in line32 and line35; inconsistency in font size of illustrations, such as Figure 6; lack of necessary explanations for the letters of the formulas, such as the equations about factual accuracy and phrase similarity.

**Questions:**

- Q1: Why only three tasks are designed and what is the motivation? Why not design judgment questions or are there any other possible tasks needed to assess the quality of scientific validity of model outputs?
- Q2: Both datasets of ClimaQA constructed in this paper are relatively small in size. What knowledge of climate is covered by these QA pairs? It shouldn't be very comprehensive, and I'm concerned that this only focuses on certain small branches of scientific knowledge. Is it possible to make some statistical analysis of the datasets to illustrate this?

---

> ### Author Response · Authors · 2024-11-24
> **Response to Reviewer mkPm - Part 1**
>
> We appreciate the constructive feedback and thoughtful suggestions. We address the concerns as follows:
>
> > One of the core innovations of the paper is the tool ClimaGen that automates the construction of ClimaQA, which involves the QA principle and prompt assembler. From line345 it appears to be a direct reference to other articles, which the authors do not describe in detail. However, this part directly affects the quality of the input prompt, which determines the quality of the generated QA pairs.
>
> We understand the importance of the QA principles and prompt assembler in ensuring the quality of the generated QA pairs. The assembled prompt, which plays a critical role in the generation process, is indeed explained in detail in Appendix 4. Hope that this provides clarity in the matter.
>
> > This paper proposes a benchmark to assess the quality and scientific validity of the output of climate base models, but there is a lack of experimental validation that the QA pairs in the benchmark are scientifically meaningful, i.e., how to prove that the constructed dataset is able to help future research on
>
> To ensure the scientific validity of the benchmark, we selected textbooks that cover a broad range of critical topics in climate science, reflecting both technical and qualitative aspects. These textbooks were chosen to provide a comprehensive understanding of the field, from foundational concepts to advanced research topics. Please refer to the newly added Appendix 1 for more details. Moreover, domain experts were involved in validating the generated QA pairs to confirm their scientific accuracy. This expert validation ensures that the dataset is scientifically meaningful and aligns with established climate science knowledge. As a result, we are confident that this dataset will contribute to advancing future research on climate foundation models, improving the quality and scientific validity of their outputs.
>
> > Lack of detailed information about the data source. The authors emphasize the data sources as 18 graduate-level textbooks, but give no detailed information about the data sources to prove it.
>
> We have now added detailed information about the data sources in Appendix 1, including the specific textbooks used for question generation. This additional information provides clarity on the selection and scope of the textbooks, helping to substantiate the validity of our dataset.
>
> > The experimental section lacks a case study. There is no visual comparison between the actual performance of the existing LLMs on these tasks and the difference between the groundtruth.
>
> Thanks for bringing this up. We have added a new case study in Appendix 2.3, where we compare and analyze the outputs of different models along with their corresponding evaluation scores. This case study provides a visual comparison between the actual performance of existing LLMs on the tasks and the ground truth where we demonstrate how the newly proposed metrics provide a reliable and robust evaluation.
>
> > Writing flaws: lack of citations for some important ideas, such as in line32 and line35; inconsistency in font size of illustrations, such as Figure 6; lack of necessary explanations for the letters of the formulas, such as the equations about factual accuracy and phrase similarity.
>
> Thank you for the feedback, we have fixed these.

---

> > ### Author Response · Authors · 2024-11-24
> > **Response to Reviewer mkPm - Part 2**
> >
> > > Why only three tasks are designed and what is the motivation? Why not design judgment questions or are there any other possible tasks needed to assess the quality of scientific validity of model outputs?
> >
> > The three tasks—MCQ, FreeformQA, and ClozeQA—were selected to cover a broad range of question types that assess different aspects of model performance. The MCQ task evaluates the decision-making ability, FreeformQA assesses the models' ability to generate detailed and accurate responses, and ClozeQA focuses on contextual understanding of scientific terminology. While we considered other complex judgemental tasks, such as causal ordering of scientific statements and factual entailment, the generator LLM could not produce valid QA pairs for this task due to its inherent complexity. We are also exploring the possibility of extending the benchmark to include multimodal tasks in the future to further enrich the evaluation of model outputs.
> >
> > >  Both datasets of ClimaQA constructed in this paper are relatively small in size. What knowledge of climate is covered by these QA pairs? It shouldn't be very comprehensive, and I'm concerned that this only focuses on certain small branches of scientific knowledge. Is it possible to make some statistical analysis of the datasets to illustrate this?
> >
> > The QA pairs in ClimaQA cover a broad range of climate science knowledge through the careful selection of textbooks that span key areas such as atmospheric science, aerosols, clouds in the climate system, geostatistics, and technical aspects of climate science. These textbooks were chosen to reflect both breadth and depth across critical topics in the field. We have included a statistical analysis in the newly added Appendix 1, which demonstrates the variety of topics covered by the QA pairs.

---

> > > ### Comment · Reviewer_mkPm · 2024-12-03
> > > **Response to the rebuttal**
> > >
> > > Thanks to the author's response, some of my concerns were resolved, so I've updated my score.

---

> > > > ### Author Response · Authors · 2024-12-03
> > > >
> > > > Thank you for reviewing the rebuttal and updating the scores!

---

### Official Review · Reviewer_wZYi · 2024-10-31

**Soundness:** 3
**Presentation:** 3
**Contribution:** 3
**Rating:** 6
**Confidence:** 3

**Summary:**

The paper presents a framework that significantly enhances the reliability and scientific rigor of AI models in climate science by generating domain-specific QA benchmarks. With ClimaQA-Gold, an expert-validated dataset, and ClimaQA-Silver, a large-scale synthetic set, the framework tests foundation models across different complexity levels, addressing factual accuracy, scientific reasoning, and hypothetical scenarios. By leveraging retrieval-augmented generation (RAG) and innovative metrics like Factual Accuracy and Phrase Similarity, ClimaGen facilitates more accurate model evaluation and fine-tuning. While advanced models like GPT-4 show promising results, they still face challenges in complex scientific questions, highlighting ClimaGen's role in driving further advancements in specialized climate AI. An expanded corpus and annotated dataset could enhance its robustness and adaptability to broader applications in scientific inquiry.

**Strengths:**

- The ClimaGen framework is tailored to climate science, which enhances the reliability of AI applications in this field by addressing unique requirements for factual rigor, domain-specific knowledge, and scientific reasoning.
- Combining an expert-validated gold-standard dataset with a large-scale synthetic dataset provides comprehensive resources for training and evaluating foundation models, supporting both high-quality benchmarking and scalability.
- The benchmark evaluates models across three question complexity levels (factual recall, reasoning, hypothetical scenarios) and multiple task types (MCQ, freeform, and cloze), which ensures a well-rounded assessment of model performance in scientific inquiry.
- Metrics like Factual Accuracy and Phrase Similarity for freeform and cloze tasks, respectively, offer a nuanced evaluation, considering both factual precision and contextual understanding.
- The use of RAG improves models' performance on reasoning questions, demonstrating a promising adaptation technique for enhancing model outputs.

**Weaknesses:**

- The framework relies on a relatively small set of graduate-level climate textbooks, which may limit the diversity of contextual information and scientific perspectives in the generated questions.
- The expert-annotated dataset (ClimaQA-Gold) is relatively small, which may restrict the scalability and generalizability of automated annotations in other scientific disciplines.
- Although improvements were noted, the models often struggle with reasoning-based questions, suggesting that further work is needed to refine reasoning capabilities in foundation models.
- The BLEU and BERTScore metrics show some bias towards the model used for question generation (GPT-3.5-turbo), which may affect the objectivity of the evaluation.

**Questions:**

Suggested Improvements:

- Increase the number and diversity of climate science sources, including a broader range of textbooks, research papers, and domain-specific databases. This would enhance the contextual diversity and robustness of the generated questions.
- Expand the expert-validated dataset to strengthen the framework's reliability and support automated annotations in additional scientific fields.
- Implement a tiered evaluation system that emphasizes model performance on reasoning and hypothetical questions separately, allowing for a deeper analysis of model strengths and weaknesses in scientific reasoning.
- Further experiment with retrieval-augmented generation using a wider array of sources or explore weighted retrieval to reduce distractors and improve information relevance.
- Introduce additional objective metrics to balance any biases introduced by BLEU or BERTScore, potentially incorporating domain-specific metrics tailored to scientific language accuracy and precision.

---

> ### Author Response · Authors · 2024-11-24
> **Response to Reviewer wZYi**
>
> We appreciate the constructive feedback and thoughtful suggestions. We address the concerns as follows:
>
> > The framework relies on a relatively small set of graduate-level climate textbooks, which may limit the diversity of contextual information and scientific perspectives in the generated questions.
>
> The QA pairs in ClimaQA cover a broad range of climate science knowledge through the careful selection of textbooks that span key areas such as atmospheric science, aerosols, clouds in the climate system, geostatistics, and technical aspects of climate science. These textbooks were chosen to reflect both breadth and depth across critical topics in the field. We have included a statistical analysis in the newly added Appendix 1, which demonstrates the distribution of topics covered by the QA pairs.
>
> > Combining an expert-validated gold-standard dataset with a large-scale synthetic dataset provides comprehensive resources for training and evaluating foundation models, supporting both high-quality benchmarking and scalability.
>
> We agree that the relatively small size of the ClimaQA-Gold dataset limits its scalability and generalizability. This is a recognized limitation, and we plan to expand the dataset and improve the annotation process in future work. We acknowledge that experiments would need to be repeated for each domain to ensure broader applicability.
>
> > Although improvements were noted, the models often struggle with reasoning-based questions, suggesting that further work is needed to refine reasoning capabilities in foundation models.
>
> We consider this observation a strength of our work, as it highlights the limitations of current foundation models. By capturing these challenges, the benchmark provides valuable insights that can guide future improvements in model reasoning capabilities.
>
> > The BLEU and BERTScore metrics show some bias towards the model used for question generation (GPT-3.5-turbo), which may affect the objectivity of the evaluation.
>
> We agree with this observation which can be explained by noting that BLEU is not immune to paraphrasing, and BERTScore does not validate scientific accuracy. This reinforces the motivation behind our proposed Factual Accuracy metric, which evaluates factual and contextual correctness for a more robust assessment. We can see from Figure 5 that Factual Accuracy does not show this bias and this is concluded in the paper. Kindly refer to the newly added case study provided in Appendix 2.3 for a detailed analysis of how the proposed metrics prove to be robust.
>
> > Implement a tiered evaluation system that emphasizes model performance on reasoning and hypothetical questions separately, allowing for a deeper analysis of model strengths and weaknesses in scientific reasoning.
>
> This is a great suggestion. We agree that implementing a tiered evaluation system focusing on reasoning and hypothetical questions separately could provide a deeper analysis of model strengths and weaknesses. This is an interesting direction for future work, and we will consider it to enhance the granularity of the evaluation.
>
> > Further experiment with retrieval-augmented generation using a wider array of sources or explore weighted retrieval to reduce distractors and improve information relevance.
>
> We agree that using a wider array of sources or weighted retrieval could help reduce distractors and improve information relevance in retrieval-augmented generation. This is a promising direction for future work to enhance the quality and reliability of generated outputs and the ClimaQA benchmark will be instrumental in evaluating these strategies.
>
> > Introduce additional objective metrics to balance any biases introduced by BLEU or BERTScore, potentially incorporating domain-specific metrics tailored to scientific language accuracy and precision.
>
> We concur that domain-specific metrics would enhance evaluation. While developing them is challenging, we see it as an exciting opportunity to refine scientific accuracy assessments with collaboration from domain experts in future work.

---

### Official Review · Reviewer_hgtP · 2024-11-02

**Soundness:** 3
**Presentation:** 3
**Contribution:** 2
**Rating:** 8
**Confidence:** 4

**Summary:**

The authors present a benchmark that they have developed for evaluating LLMs on question-answer tasks specialized for the domain of climate science. They developed two datasets: ClimaQA-Gold containing 502 questions that have been annotated by climate science experts, and ClimaQA-Silver containing 3000 questions generated by their framework for LLM-powered generation of question-answered pairs based on a corpus of graduate-level textbooks. Their benchmark contains three types of questions: "free form" which expects the answers to be free-form text, "multiple choice" where the answer is given as an indicator to one of the proposed answer choices, and "cloze" which contains certain parts of text marked as "blank" and the task is to find the correct words to fill in the blanks. Questions come in three levels of difficulty: "base" which focuses on simple information retrieval, "reasoning" which requires the connection of multiple facts to form a conclusion, and "hypothetical scenario" which requires the application of knowledge to novel contexts. They conduct an empirical evaluation of several popular LLMs (including extensions such as RAG). The scores that the models achieve form a wide range which demonstrates the utility of the benchmark for evaluating LLMs on QA tasks for the specific domain of choice.

**Strengths:**

**(S1)** The proposed benchmark represents a significant step in creating a rigorous evaluation for QA models on climate-specific knowledge and reasoning tasks.

**(S2)** The authors provide both a human-curated corpus for final evaluation and an automatically augmented corpus for fine-tuning.

**(S3)** The fact that the benchmark is able to stratify models based on their performance on this task demonstrates that it is challenging, which will help it become a reference point for future models that are built specifically for the climate science domain.

**Weaknesses:**

**(W1)** The domain that the authors focus on can be deemed as a very niche one. The authors could have tried to do a better job at providing concrete arguments in the introduction that underline the significance of this domain. Specifically, it would be good to explain why we need LLMs to be good at this specific domain. If they become better at this domain, which applications would this unlock?

**(W2)** There are certain small issues with the presentation: (1) the authors use the term "foundation models" but probably just want to say either "large language models" or "question-answering models". Foundation models are pre-trained on large corpora and used mainly for transfer learning across different tasks, but many of them are not LLMs. It would help to use a more specific term. (2) Formulas presented in section 3 use some symbols that were not explicitly defined (e.g. $p_s$, $p_r$). (3) The numbers in Figure 4 seem to keep going up (albeit at a reducing pace), and yet the authors conclude that a "context window of size 4 most effectively differentiates between correct and incorrect answers". From the figure, it is unclear to me why this context window is optimal.

**(W3)** This is a minor weakness, but the provided corpus is relatively modest, compared to other ones. I don't think this is too much of an issue given that it is probably large enough to sufficiently evaluate a QA model, and a small size makes it more manageable. That said, it would be good to provide some sense of how well the provided corpus covers the actual domain (e.g. by giving the readers a sense of the subdomains and the number of questions from each subdomain).

**Questions:**

**(Q1)** In section 5.2 you mention that there are 5 books used to derive the corpus of questions, and an additional 13 books that are used for the RAG model as a retrieval corpus. Do we know if all relevant information between these two sets of books overlaps? In other words, do we know if the information in the 13 books is sufficient to support all the questions that were produced by using the 5 books?

**(Q2)** In the introduction you mention "Creation of publicly releasable datasets". Could you please clarify what you mean by a "releasable" dataset?

---

> ### Author Response · Authors · 2024-11-24
> **Response to Reviewer hgtP**
>
> We appreciate the constructive feedback and thoughtful suggestions. We address the concerns as follows:
>
> > The domain that the authors focus on can be deemed as a very niche one. The authors could have tried to do a better job at providing concrete arguments in the introduction that underline the significance of this domain. Specifically, it would be good to explain why we need LLMs to be good at this specific domain. If they become better at this domain, which applications would this unlock?
>
> Climate change is one of the most pressing global challenges today, with profound impacts on ecosystems, economies, and societies. Ensuring that LLMs perform well in this domain can unlock applications in climate policy analysis, environmental decision-making, and public education. By improving LLMs' understanding of climate science, we can empower stakeholders to make informed decisions, develop actionable solutions, and foster broader awareness of climate issues. We have modified the introduction to reflect this strongly.
>
>
> > (1) the authors use the term "foundation models" but probably just want to say either "large language models" or "question-answering models". Foundation models are pre-trained on large corpora and used mainly for transfer learning across different tasks, but many of them are not LLMs. It would help to use a more specific term.
>
> Modified the occurrences accordingly
>
> > (2) Formulas presented in section 3 use some symbols that were not explicitly defined (e.g. ps, pr).
>
> Thanks for the feedback. These are fixed.
>
> > The numbers in Figure 4 seem to keep going up (albeit at a reducing pace), and yet the authors conclude that a "context window of size 4 most effectively differentiates between correct and incorrect answers". From the figure, it is unclear to me why this context window is optimal.
>
> The choice of a context window size of 4 is based on its ability to yield the largest difference in mean cosine similarities between correct and incorrect answers while still maintaining sufficiently high scores for correct answers. That said, we acknowledge that this conclusion is empirical and may vary depending on the specific dataset and domain.
>
> > This is a minor weakness, but the provided corpus is relatively modest, compared to other ones. I don't think this is too much of an issue given that it is probably large enough to sufficiently evaluate a QA model, and a small size makes it more manageable. That said, it would be good to provide some sense of how well the provided corpus covers the actual domain (e.g. by giving the readers a sense of the subdomains and the number of questions from each subdomain).
>
> The QA pairs in ClimaQA cover a broad range of climate science knowledge through the careful selection of textbooks that span key areas such as atmospheric science, aerosols, clouds in the climate system, geostatistics, and technical aspects of climate science. These textbooks were chosen to reflect both breadth and depth across critical topics in the field. We have included a statistical analysis in the newly added Appendix 1, which demonstrates the distribution of topics covered by the QA pairs.
>
> > In section 5.2 you mention that there are 5 books used to derive the corpus of questions, and an additional 13 books that are used for the RAG model as a retrieval corpus. Do we know if all relevant information between these two sets of books overlaps? In other words, do we know if the information in the 13 books is sufficient to support all the questions that were produced by using the 5 books?
>
> There is very little overlap between the information in the 5 books used for question generation and the 13 books used as the retrieval corpus. This is reflected in the retrieval-augmented generation (RAG) scores, as some questions lack sufficient supporting information in the retrieval corpus. Addressing this is part of our future work, where we plan to expand and refine the retrieval corpus to ensure better alignment with the question-generation sources.
>
> > In the introduction you mention "Creation of publicly releasable datasets". Could you please clarify what you mean by a "releasable" dataset?
>
> By "releasable" dataset, we mean that the dataset will be properly cleaned and organized, and made publicly available for use. Currently, it is provided as supplementary files, and we plan to release it online and make it easy to access.

---

> ### Comment · Reviewer_hgtP · 2024-12-02
> **Response to Rebuttal**
>
> I very much appreciate the authors' thorough responses and provide any additional comments below:
>
> * (W1) - The update to the intro that the authors have made helps bolster the motivation.
>
> * (W2) (1 and 2) Thank you for making the fixes! (3) I appreciate the clarification but suggest that it would be better if these were plotted in the figure in order to make the tradeoff more clear.
>
> * (W3) I appreciate the explanation and the breakdown provided by the authors!
>
> * (Q1) I appreciate the explanation.
>
> * (Q2) Thank you for clarifying!
>
> Finally, I have updated my score a little but in support of this paper getting accepted.

---

> > ### Author Response · Authors · 2024-12-03
> >
> > Thank you for reviewing the rebuttal and updating the scores!

---

### Official Review · Reviewer_QKkY · 2024-11-03

**Soundness:** 3
**Presentation:** 3
**Contribution:** 3
**Rating:** 6
**Confidence:** 4

**Summary:**

The authors present ClimaQA, a benchmark for language models, assessing skills in the area of climate science. The research introduces an automated pipeline (ClimaGen) to generate question-answer pairs from graduate textbooks, resulting in two datasets: ClimaQA-Gold (expert-validated) and ClimaQA-Silver (synthetic). The article presents three types of tasks: multiple-choice, freeform, and cloze with varying levels of complexity to challenge models on factual recall, scientific reasoning, and scenario application. For evaluation, the research develops dedicated metrics for freeform and cloze tasks and employs a fine-tuned LLM for the synthetic annotation process using expert-annotated data. The study evaluates several state-of-the-art large language models on the created benchmark under different settings, such as default configurations, few-shot prompting, and retrieval-augmented generation.

**Strengths:**

- The research addresses a lack of expert-annotated benchmarks in climate science and proposes a novel algorithmic framework to accelerate scientific benchmark creation through synthetic generation and expert validation.
- The synthetic scaling approach enables continuous updates to the benchmark, accommodating new findings in climate science research.
- The benchmark's incorporation of three distinct tasks (multiple-choice questions, freeform, and cloze) reduces potential biases arising from models' familiarity with specific question formats, unlike benchmarks limited to a single task type.
- The implementation of three well-defined complexity levels enables a comprehensive assessment of models’ knowledge and application capabilities, allowing precise identification of areas for improvement and a better understanding of current model performance.

**Weaknesses:**

- Exclusive use of proprietary models (GPT3.5 for question generation and GPT4o-mini for freeform judgment) introduces potential bias favoring models from the same provider or those trained/fine-tuned on similar model outputs.
- The proposed metrics demonstrate an inconsistent correlation with expected language model performance. For instance, GPT3.5 exhibits superior or comparable performance against newer models, including those from the same family (GPT4o), on BLEU and BERTScore metrics as shown in Figure 5, highlighting potential metric reliability concerns in model evaluation.
- The research reports an 85% validity rate for generated multiple-choice questions in ClimaQA-Silver (Section 4.4), which may compromise the effectiveness of the evaluation. As several models on ClimaQA-Gold reach or exceed this performance level, the synthetic dataset may not provide reliable performance metrics. A comparative analysis similar to Table 3 for the synthetic dataset could illustrate this limitation.
- The study lacks quantitative comparisons with existing climate benchmarks, such as Climate Crisis or SciQAG-24D.
- The benchmark's reliance on textbook knowledge may introduce distribution bias in the evaluation process.
- The research provides a limited analysis of the expert validation process and common question generation errors, which could offer valuable insights into potential biases in the synthetic benchmark version.

**Questions:**

- Could you provide a quantitative comparison with established climate benchmarks, specifically Climate Crisis and SciQAG-24D?
- Would it be possible to present a comparative performance analysis, similar to Table 3, for the ClimaQA-Silver dataset to better understand potential quality differences between gold and silver datasets?
- Has the research considered implementing a mixture-of-experts approach for the freeform evaluation process instead of relying solely on GPT4o-mini?
- What were the selection criteria for the 18 textbooks, and how were they assessed for their relevance to recent advances in climate science?
- What methodology was used to ensure that the set of 5 textbooks provided a balanced coverage of climate science topics?

---

> ### Author Response · Authors · 2024-11-24
> **Response to Reviewer QKkY - Part 1**
>
> We appreciate the constructive feedback and thoughtful suggestions. We address the concerns as follows:
>
> > Exclusive use of proprietary models (GPT3.5 for question generation and GPT4o-mini for freeform judgment) introduces potential bias favoring models from the same provider or those trained/fine-tuned on similar model outputs.
>
> We acknowledge the concern regarding the exclusive use of proprietary models. Our choice was driven by the remarkable generalization capabilities of OpenAI models, which have consistently demonstrated strong performance across diverse domains and tasks without extensive domain-specific fine-tuning. While this introduces reliance on a single provider, it allows us to focus on the inherent strengths of these models in handling varied tasks. Future work can incorporate a broader range of models to address potential biases.
>
> > The proposed metrics demonstrate an inconsistent correlation with expected language model performance. For instance, GPT3.5 exhibits superior or comparable performance against newer models, including those from the same family (GPT4o), on BLEU and BERTScore metrics as shown in Figure 5, highlighting potential metric reliability concerns in model evaluation.
>
> We agree that BLEU and BERTScore may favor the generator model (GPT-3.5) due to the reference answers being generated by it, which could explain the observed performance discrepancies. However, BLEU is not immune to paraphrasing, and BERTScore does not validate scientific accuracy. This reinforces the motivation behind our proposed Factual Accuracy metric, which evaluates factual and contextual correctness for a more robust assessment. We can see from Figure 5 that Factual Accuracy does not show this bias and this is concluded in the paper.  Kindly refer to the newly added case study provided in Appendix 2.3 for a detailed analysis of how the proposed metric proves to be robust.
>
>
> > The benchmark's reliance on textbook knowledge may introduce distribution bias in the evaluation process.
>
> We believe that reliance on textbook knowledge is a strength rather than a limitation, as textbooks represent foundational knowledge, consensus among experts, and strong coverage of basic concepts. This ensures that our benchmark evaluates models on well-established and widely accepted information, providing a stable and reliable basis for assessment.
>
> > The research provides a limited analysis of the expert validation process and common question generation errors, which could offer valuable insights into potential biases in the synthetic benchmark version.
>
> Thank you for your valuable feedback. We have conducted further analysis of the expert validation and included more details in the newly added Appendix 5.1. During the validation of the ClimaQA-Gold dataset, experts identified several common reasons for rejecting generated QA pairs, including *wrong answer, bad context, question doesn't make sense, not enough info,* and other factors. While most of these errors are attributed to limitations in the generator LLM, the bad context category highlights issues with the seed context, suggesting that improvements in preprocessing could help reduce such errors. This is an area we plan to address in future work to enhance the quality and reliability of the benchmark.

---

> > ### Author Response · Authors · 2024-11-24
> > **Response to Reviewer QKkY - Part 2**
> >
> > > Could you provide a quantitative comparison with established climate benchmarks, specifically Climate Crisis and SciQAG-24D?
> >
> > Could you elaborate on what is meant by quantitative comparison here?
> >
> > > Would it be possible to present a comparative performance analysis, similar to Table 3, for the ClimaQA-Silver dataset to better understand potential quality differences between gold and silver datasets?
> >
> > Thank you for your suggestion. We have now included a comparative performance analysis for the ClimaQA-Silver dataset in Appendix 2.2. We assessed the models in their default settings on a subset of the ClimaQA-Silver dataset, consisting of 200 questions for each task type. The column-wise trends observed in the results are largely consistent with those from the ClimaQA-Gold dataset. However, a notable difference lies in the relative difficulty of the complexities in the MCQ task - Reasoning questions have better scores than the Base questions, even though the overall scores do not vary significantly. This discrepancy highlights the intricate nature of the MCQ generation process. Additionally, this variation may also be attributed to the relatively small number of questions in each column. Future work should focus on scaling up the expert validation process to enhance the quality of the automated annotation pipeline, thereby addressing these challenges and improving overall silver-dataset reliability.
> >
> >
> > > Has the research considered implementing a mixture-of-experts approach for the freeform evaluation process instead of relying solely on GPT4o-mini?
> >
> > Our evaluation strategy is not LLM-dependent and can be easily replaced with more recent models or a mixture-of-experts approach in the future. We chose GPT-4o-mini specifically for its generalizability and strong performance in zero-shot factual entailment in the Climate Fever dataset. This made it a suitable choice for evaluating freeform responses. However, we recognize that other models or approaches, including mixture-of-experts, could potentially offer further improvements, and this is an area we will consider exploring in future work.
> >
> > > What were the selection criteria for the 18 textbooks, and how were they assessed for their relevance to recent advances in climate science?
> >
> > The textbooks were chosen from the virtual bookshelf of a professor in atmospheric physics and broadly represent a mixture of graduate and expert textbooks in the physical climate, with a particular focus on the role of aerosol in the climate system (which provides one of the key uncertainties in climate projections). Please refer to the newly added Appendix 1 for more details.
> >
> > > What methodology was used to ensure that the set of 5 textbooks provided a balanced coverage of climate science topics?
> >
> > The QA pairs in ClimaQA cover a broad range of climate science knowledge through the careful selection of textbooks that span key areas such as atmospheric science, aerosols, clouds in the climate system, geostatistics, and technical aspects of climate science. These textbooks were chosen to reflect both breadth and depth across critical topics in the field. We have included a statistical analysis in the newly added Appendix 1, which demonstrates the distribution of topics covered by the QA pairs.

---

> > > ### Comment · Reviewer_QKkY · 2024-12-02
> > >
> > > >> Could you provide a quantitative comparison with established climate benchmarks, specifically Climate Crisis and SciQAG-24D?
> > >
> > > > Could you elaborate on what is meant by quantitative comparison here?
> > >
> > > A table with numerical results of different models between introduced benchmark, and existing ones. This could show whether it's harder to answer than previous work.
> > >
> > > I appreciate your efforts in improving the manuscript and have increased my score.

---

> > > > ### Author Response · Authors · 2024-12-03
> > > >
> > > > Thanks for reviewing the rebuttal and updating the scores!
> > > >
> > > > We agree that a comparative analysis with other benchmarks could provide more insights although merely comparing scores may not be the best way to do it. We plan to address this in the future work.

---

### Official Review · Reviewer_vgwR · 2024-11-05

**Soundness:** 3
**Presentation:** 3
**Contribution:** 4
**Rating:** 8
**Confidence:** 3

**Summary:**

The authors introduce ClimaQA, a question-answering benchmark designed to evaluate climate foundation models in handling scientific inquiries. This paper develops ClimaGen, a framework for generating question-answer pairs with climate experts in the loop, resulting in two datasets: ClimaQA-Gold (expert-validated) and ClimaQA-Silver (synthetic). The benchmark assesses models across multiple task forms (MCQ, freeform, and cloze) to gauge factual accuracy and scientific reasoning. The paper presents an analysis of several LLMs on ClimaQA.

**Strengths:**

· Originality: ClimaQA stands out as a pioneering benchmark specifically tailored for climate science. The framework’s approach of combining domain expertise with synthetic QA generation represents an innovative solution to the lack of reliable climate QA datasets.

· Quality: The benchmark’s rigorous development process, including the expert validation of ClimaQA-Gold and systematic generation of ClimaQA-Silver, ensures high-quality question-answer pairs that reflect real-world scientific standards. The choice of using diverse question formats (MCQ, freeform, cloze) is well-aligned with testing scientific reasoning skills.

· Clarity: The paper is organized and highly readable, with clear descriptions of each dataset and evaluation metric. The figures illustrate the QA generation and annotation processes well, making the methodology transparent.

**Weaknesses:**

· Complexity of Generated Questions: While the QA generation pipeline is well-designed, some questions in ClimaQA-Silver may lack the complexity required to thoroughly test advanced climate models. Supplementing the dataset with more nuanced, multi-layered questions could improve its utility.

· Dependence on Manual Annotation for ClimaQA-Gold: Although ClimaQA-Gold is an expertly annotated dataset, its reliance on manual validation limits scalability. Developing a semi-automated validation process or implementing peer review from multiple experts could reduce bias and improve reproducibility.

**Questions:**

1. What is the potential for scaling the ClimaQA benchmark to other scientific domains? Could ClimaGen be adapted for other domains like environmental science or epidemiology, and what would be the main challenges?

2. Have you considered additional metrics for freeform QA evaluation? While factual accuracy is addressed, additional metrics like coherence and logical flow might further enhance the assessment of model responses on freeform tasks.

---

> ### Author Response · Authors · 2024-11-24
> **Response to Reviewer vgwR**
>
> We appreciate the constructive feedback and thoughtful suggestions. We address the concerns as follows:
>
> >  Complexity of Generated Questions: While the QA generation pipeline is well-designed, some questions in ClimaQA-Silver may lack the complexity required to thoroughly test advanced climate models. Supplementing the dataset with more nuanced, multi-layered questions could improve its utility.
>
> We agree that creating a dataset with more nuanced, multi-layered questions is a valuable suggestion, and we will consider it to enhance the dataset's ability to better evaluate model performance in future iterations.
>
> > Dependence on Manual Annotation for ClimaQA-Gold: Although ClimaQA-Gold is an expertly annotated dataset, its reliance on manual validation limits scalability. Developing a semi-automated validation process or implementing peer review from multiple experts could reduce bias and improve reproducibility.
>
> This is an insightful suggestion. We agree that using multiple experts in the validation process is a valuable approach to reduce bias and improve reproducibility. Implementing a semi-automated validation process involving peer review from multiple experts is an excellent idea and something we would love to explore in future work.
>
> > What is the potential for scaling the ClimaQA benchmark to other scientific domains? Could ClimaGen be adapted for other domains like environmental science or epidemiology, and what would be the main challenges?
>
> The ClimaQA benchmark has significant potential for scaling to other scientific domains, as long as we have access to relevant data sources. The main challenges in adapting ClimaGen for these domains would be collecting appropriate data sources and securing domain experts for validation. Once these resources are in place, the framework could be effectively extended to evaluate model performance across different scientific fields.
>
> > Have you considered additional metrics for freeform QA evaluation? While factual accuracy is addressed, additional metrics like coherence and logical flow might further enhance the assessment of model responses on freeform tasks.
>
> We agree that metrics like coherence and logical flow would enhance freeform QA evaluation. In addition, we are considering a tiered evaluation system for a more nuanced analysis. These are promising directions for future work, and we plan to explore them further.

---

### Author Response · Authors · 2024-11-24
**Overarching Summary**

We appreciate the insightful reviews and thoughtful suggestions from the reviewers, which have significantly helped us in improving our work.

We are pleased that the following aspects of our work were appreciated:
- **Addressing a Critical Gap with Expert Validation:** This research addresses the lack of expert-annotated benchmarks in climate science by proposing a novel algorithmic framework that accelerates benchmark creation through synthetic generation and expert validation. [vgwR, QKkY, hgtP, wZYi, mkPm]
- **Diverse and Rigorous Evaluation:** Incorporates multiple question types (MCQ, freeform, cloze) and complexity levels (factual recall, reasoning, hypothetical), ensuring comprehensive model assessment. The ability to stratify models based on task performance highlights the benchmark's challenge, positioning it as a reference for future climate science models. [vgwR, QKkY, hgtP, wZYi, mkPm]
- **Nuanced Evaluation Metrics:** Uses metrics like Factual Accuracy and Phrase Similarity to provide a detailed evaluation of model performance, considering both factual precision and contextual understanding. [wZYi]

Below, we outline the key changes made in the revised version of our paper. To facilitate review, all changes have been highlighted in red:

- **Revised terminology:** Replaced "foundation models" with "LLMs" or "question-answering models" to better reflect our focus and enhanced the introduction with a clearer motivation for improving LLMs in climate science. [hgtP]
- **Expanded data source details:** Added information about the textbook sources and a statistical analysis of question coverage (Appendix 1). [QKkY, hgtP, wZYi, mkPm]
- **New case study:** Included an analysis of state-of-the-art LLM performance on benchmark questions, along with evaluation scores, to demonstrate the robustness of the proposed metrics (Appendix 2.3). [mkPm]
- **Silver vs. Gold comparison:** Assessed models on the silver dataset, analyzing differences from the gold dataset to highlight potential limitations of automated annotation (Appendix 2.2). [QKkY]
- **QA generation error analysis:** Added a statistical breakdown of reasons for scientifically inaccurate generated QA pairs identified by domain experts, offering insights into pipeline improvement (Appendix 5.1). [QKkY]
- **Presentation fixes:** Addressed minor issues such as symbol descriptions and font sizes in figures. [hgpt, mkpm]
- **Dataset Expansion:** Expanded the ClimaQA-Gold MCQ dataset by incorporating expert-validated questions added after the paper's initial submission, re-ran the experiments, and updated the results and supplementary materials accordingly.
- **Additional Model Evaluation:** Conducted an evaluation of the Gemini-1.5-Flash model to investigate the impact of long-context prompting.

We believe these revisions have addressed the reviewers' concerns and enhanced our work. Thank you for your valuable feedback and recognition.

*Note:* We modified the title in the PDF but could not change it in the portal itself.

---

### Meta-Review · Area_Chair_gGa9 · 2024-12-20

**Metareview:**

This well-written paper has been evaluated by 5 knowledgeable reviewers. They have unanimously recommended its acceptance (including 2 straight acceptances and 3 marginal acceptances). The authors have provided a rebuttal, and it helped the reviewers reassess their scores. The maturity of the presented work and its scope are appropriate for publication at ICLR.

**Additional Comments On Reviewer Discussion:**

No discussion was needed, the reviewers predominantly agreed on this paper and its relative strengths and non-critical weaknesses.

---

### Decision · Program_Chairs · 2025-01-22

Accept (Poster)